# GRIN2B disease-associated mutations disrupt the function of BK channels and NMDA receptor signaling nanodomains

Rebeca Martínez-Lázaro[1,2] , Teresa Minguez-Viñas[1,2] , Andrea Reyes-Carrión[1,2] , Ricardo Gómez[1,2] , Diego Alvarez de la Rosa[1,2] , David Bartolomé-Martín[2,3] , and Teresa Giraldez[1,2]

**Large conductance calcium-activated potassium channels (BK channels) are unique in their ability to respond to two distinct physiological stimuli: intracellular Ca²⁺ and membrane depolarization. In neurons, these channels are activated through a coordinated response to both signals; however, for BK channels to respond to physiological voltage changes, elevated concentrations of intracellular Ca²⁺ (ranging from 1 to 10 µM) are necessary. In many physiological contexts, BK channels are typically localized within nanodomains near Ca²⁺ sources (∼20–50 nm), such as N-methyl-D-aspartate receptors (NMDARs; encoded by the GRIN genes). Since the direct evidence of NMDAR–BK channel coupling reported by Isaacson and Murphy in 2001 in the olfactory bulb, further studies have identified functional coupling between NMDARs and BK channels in other regions of the brain, emphasizing their importance in neuronal function. Mutations in the genes encoding NMDAR subunits have been directly linked to developmental encephalopathies, including intellectual disability, epilepsy, and autism spectrum features. Specifically, mutations V15M and V618G in the GRIN2B gene, which encodes the GluN2B subunit of NMDARs, are implicated in the pathogenesis of GRIN2B-related neurodevelopmental disorders. Here, we explored the effects of these two GluN2B mutations on NMDAR–BK channel coupling, employing a combination of electrophysiological, biochemical, and imaging techniques. Taken together, our results demonstrate that mutation V618G specifically disrupts NMDAR–BK complex formation, impairing functional coupling, in spite of robust individual channel expression in the membrane. These results provide a potential mechanistic basis for GRIN2B-related pathophysiology and uncover new clues about NMDAR–BK complex formation.**

## Introduction

Large conductance Ca²⁺- and voltage-activated K⁺ channels (KCa1.1, BK, MaxiK, or slo1) are expressed in cellular membranes as homotetramers of α subunits encoded by the KCNMA1 (Slo1) gene (Cui et al., 2009; Latorre et al., 2017). BK channels have a wide range of specialized physiological functions across various excitable and non-excitable tissues, such as muscle, kidney, gastrointestinal tract, salivary glands, and bone (Echeverría et al., 2024). In the central nervous system, they are found in the olfactory system, neocortex, basal ganglia, hippocampus, and thalamus (Trimmer, 2015; Kshatri et al., 2018). Within excitable cells, BK channels are primarily involved in shaping the action potential and regulating firing frequency, as well as neurotransmitter release (Storm, 1987; Bean, 2007; Contet et al., 2016). In addition, they have been shown to regulate synaptic transmission and plasticity (Isaacson and Murphy, 2001; Zhang et al., 2018; Gómez et al., 2021).

A key physiological characteristic of BK channels is the dependence of simultaneous membrane depolarization and increased intracellular Ca²⁺ for activation (Marty, 1981; Pallotta et al., 1981). In many cell types, BK channel activation relies on localized Ca²⁺ rises that reach micromolar concentrations, significantly higher than the typical resting cytosolic levels of 100–300 nM (Fakler and Adelman, 2008). Physiologically, BK channels are often situated near other proteins that serve as intracellular Ca²⁺ sources, functioning within specialized Ca²⁺ nano- or micro-domains (Shah et al., 2021; Gonzalez-Hernandez et al., 2023). In the postsynapse, the association of BK to N-methyl-D-aspartate receptors (NMDARs) in Ca²⁺ nanodomains has been proposed to be involved in regulation of synaptic transmission and plasticity (Isaacson and Murphy, 2001; Zhang et al., 2018; Gómez et al., 2021; Tazerart et al., 2022). NMDARs, encoded by the GRIN genes, are

[1]Departamento de Ciencias Médicas Básicas-Fisiología, Universidad de La Laguna, Tenerife, Spain; [2]Instituto de Tecnologías Biomédicas, Universidad de La Laguna, Tenerife, Spain; [3]Departamento de Bioquímica y Biología Molecular, Universidad de La Laguna, Tenerife, Spain.

Correspondence to Teresa Giraldez: giraldez@ull.edu.es; David Bartolomé-Martín: dbartolo@ull.edu.es.

heterotetrameric ligand-gated ion channels that belong to the family of ionotropic glutamate receptors, together with the α-amino-3-hydroxy-5-methyl-4-isoxazolepropionic acid (AMPAR) and kainate receptors (Reiner and Levitz, 2018; Hansen et al., 2021). In physiological conditions these receptors mediate the inflow of $Na^+$ and $Ca^{2+}$ and outflow of $K^+$ (Hansen et al., 2018). NMDARs facilitate the gradual entry of $Ca^{2+}$ into cells in response to neuronal stimuli, making them essential for processes like synaptic plasticity, learning, memory, and other advanced cognitive functions (Paoletti et al., 2013). Their physiological importance is highlighted by the association between NMDAR dysfunction and a range of neurological and psychiatric disorders, such as Alzheimer's (Mota et al., 2014) and Huntington's diseases (Fernandes and Raymond, 2009), schizophrenia, and stroke (Paoletti et al., 2013), as well as major depressive disorder (Molero et al., 2018).

NMDARs are expressed at the neuronal membrane as tetramers of various subunit combinations. Seven homologous NMDAR subunits have been identified: the essential GluN1/NR1 subunit (encoded by GRIN1), four GluN2/NR2 subunits (GluN2A, GluN2B, GluN2C, and GluN2D; encoded by GRIN2A, GRIN2B, GRIN2C, and GRIN2C, respectively), and two GluN3/NR3 subunits (GluN3A and GluN3B; encoded by GRIN3A and GRIN3B). The GluN1 and GluN3 subunits have binding sites for co-agonists, such as glycine or D-serine, while the GluN2 subunits feature a binding site for the agonist glutamate (Paoletti et al., 2013). In the adult brain, the most common configuration of NMDARs includes two GluN1 combined with two either GluN2A or GluN2B subunits (GluN1/GluN2) (Hansen et al., 2018). NMDARs are highly responsive to glutamate, with a half-maximal effective concentration in the micromolar range, and they undergo voltage-dependent blockade by $Mg^{2+}$ ions (Mayer et al., 1984; Nowak et al., 1984). Their slow gating kinetics (Lester et al., 1990) and notable $Ca^{2+}$ permeability (MacDermott et al., 1986; Mayer and Westbrook, 1987) allow postsynaptic NMDARs to effectively sense and interpret the simultaneous activity of both presynaptic and postsynaptic neurons. Specifically, glutamate released from the presynaptic neuron binds to the receptor, while depolarization of the postsynaptic membrane via AMPARs alleviates the $Mg^{2+}$ block. This coordination activates the NMDARs, enabling $Ca^{2+}$ influx through the channel and initiating signaling cascades that can influence synaptic plasticity (Paoletti et al., 2013).

The discovery that $Ca^{2+}$ influx through glutamate-activated NMDARs could activate BK outward currents was directly demonstrated in the olfactory bulb (Isaacson and Murphy, 2001). Subsequent research by Zhang and colleagues indicated that this functional relationship might also occur in other brain regions, as evidenced by the co-immunoprecipitation of BK and NMDAR in the hippocampus, cortex, cerebellum, striatum, and thalamus (Zhang et al., 2018). A role of these complexes has been reported in cortical neurons (Hayashi et al., 2016; Gómez et al., 2021; Tazerart et al., 2022), hippocampal neurons (Zhang et al., 2008; Zhang et al., 2010), nucleus accumbens (Ji et al., 2015), dorsal cochlear nucleus, (He et al., 2014) and superficial dorsal horn neurons of the thoracolumbar spinal cord (Fan et al., 2021). Function of these associations has been tested by performing

whole-cell patch-clamp recordings after applying glutamate or NMDA either toward the neuronal soma (Hayashi et al., 2016; Zhang et al., 2018) or at dendritic locations (Gómez et al., 2021; Tazerart et al., 2022). Notably, the activation of BK by NMDARs in dendrites has been shown in layer 5 pyramidal neurons from the cortex (Gómez et al., 2021; Tazerart et al., 2022; Mitchell et al., 2023), and more recently in the amygdala as well as in the CA3 region of the hippocampus (Reyes-Carrión et al., 2023). NMDAR–BK complexes have been described in both extrasynaptic and postsynaptic terminals (Isaacson and Murphy, 2001; Zhang et al., 2018; Gómez et al., 2021; Tazerart et al., 2022; Mitchell et al., 2023). Within these nanodomains, $Ca^{2+}$ entry through activated NMDARs opens BK channels, resulting in the hyperpolarization of the adjacent plasma membrane and closure of NMDAR channels by restoring $Mg^{2+}$ block. Because BK channel activation blunts NMDAR-mediated excitatory responses, it provides a negative feedback mechanism that may modulate excitability, synaptic transmission, or plasticity, depending on the location of these associations within the neuron (Gómez et al., 2021; Shah et al., 2021). Although the mechanisms underlying the association between BK and NMDAR in the nanodomains remain largely unexplored, it has been shown that the isolated GluN1 cytosolic regions directly interact in vitro with a synthesized peptide of the BKα S0–S1 loop region. In addition, the NMDAR–BK interaction is competitively diminished by a synthesized peptide from the BKα S0–S1 loop (Zhang et al., 2018). The role of the GluN2 subunits in regulating the formation or function of NMDAR–BK nanodomains remained unexplored.

In recent years, different studies have discovered inherited and de novo mutations in the GRIN genes, encoding NMDAR, that are directly related to neurodevelopmental disorders, such as mental retardation, intellectual disability, epilepsy, and autism spectrum disorders (Benke et al., 2021). Pathogenic de novo variants on the GRIN2B gene, which encodes the GluN2B subunit of NMDARs, have been linked to developmental delay and intellectual disability of variable severity with (OMIM accession no. 613970) or without (OMIM accession no. 616139) early onset seizures (Lemke et al., 2014; Swanger et al., 2016; Platzer et al., 2017; Platzer and Lemke, 2018; Benke et al., 2021). Phenotypes of GRIN2B-related disorders are highly variable among patients, ranging from mild intellectual disability without seizures to more severe encephalopathy. Additional features may include hypotonia, abnormal movements (such as dystonia), and autistic traits. We hypothesized that, based on the growing evidence that NMDAR–BK associations play a relevant role in many neuronal types, regulating synaptic function, some of these disease-related mutations may result in alterations of NMDAR–BK functional associations. In this work, we have focused on studying the effect of mutations directly linked to GRIN2B neurodevelopmental disorders on NMDAR–BK associations. Using a combination of electrophysiology and $Ca^{2+}$ imaging, molecular biology and protein biochemistry, total internal reflection microscopy, and superresolution microscopy, we now show that two disease-related mutations in the GluN2B subunit (V15M and V618G) alter the functional association of NMDAR and BK in nanodomains through different mechanisms. The

V15M mutation does not directly impact the efficiency of NMDAR–BK coupling, but it significantly reduces the membrane levels of NMDARs. This leads to fewer functional NMDAR–BK associations and may alter the molecular ratio of BK to NMDAR within the complexes. In contrast, the V618G mutation specifically affects the efficiency of NMDAR–BK coupling by changing the size of the NMDAR–BK nanodomains, likely through modifications in their molecular interactions. Notably, our data indicate that the formation of functional NMDAR–BK macrocomplexes may not solely depend on GluN1–BK interactions, as previously suggested (Zhang et al., 2018), but also involves the GluN2 subunit. Our findings with disease-related mutations indicate that the function of NMDAR–BK nanodomains is influenced by cluster size and possibly the molecular ratio of NMDAR to BK channels, rather than the distance between proteins within the nanodomain. This suggests a potential mechanism underlying the functional variability of NMDAR–BK macrocomplexes.

## Materials and methods

### Cell culture, transfection, cDNA constructs, and mutagenesis
HEK293T cells (American Type Culture Collection no. CRL-3216) were grown in Dulbecco's modified Eagle's medium (Sigma-Aldrich) supplemented with 10% FBS (Sigma-Aldrich), 1% penicillin-streptomycin (Thermo Fisher Scientific), and Mycozap (Lonza). Cells used for imaging and electrophysiology experiments were seeded on polylysine-treated glass coverslips to promote cell attachment. Cells grown to 60–80% confluence in OptiMEM culture medium were transfected with the indicated plasmid combinations using jetPRIME reagent (Polyplus) following the manufacturer's instructions. 4 h after transfection, the medium was replaced with the same medium complemented with 200 µM DL-2-amino-5-phosphonovaleric acid and 20 µM 5,7-dichlorokynurenic acid, and cells were incubated at 37°C for 24–48 h. The following plasmids were used for transient cell transfections: pEYFP-GluN1a, encoding the NMDAR-GluN1 subunit tagged with an enhanced YEP (plasmid #17928; Addgene [Luo et al., 2002]); pEGFP-GluN2B, encoding the NMDAR-GluN2B subunit tagged with an enhanced GFP (EGFP) (plasmid #17925; Addgene [Luo et al., 2002]); pEGFP-NR2B mutants, encoding for NR2B mutant subunits tagged with an EGFP (this work); pBNJ_hsloTAG, encoding for BKα subunits tagged with a DYKDDDKD flag (TAG) (Giraldez et al., 2005); pcDNA3-EGFP, encoding for an EGFP (plasmid #13031; Addgene); and Lyn-R-GECO1 (gift from Won Do Heo, Kaist Institute, Korea [plasmid # 120410; Addgene] [Kim et al., 2016]). All fluorescently tagged NMDAR plasmids were a gift from Stefano Vicini, Georgetown University School of Medicine, Washington, DC, USA (Vicini et al., 1998; Gómez et al., 2021). The transfection ratio of GluN1:GluN2B was, in most of the experiments, 1:3. When cotransfected with BK channels, the BK:GluN1:GluN2B transfection ratio was 1:1:3, except for superresolution imaging experiments in which the transfection ratio was 1:1:2.

### Proximity ligation assay
Proximity ligation assay (PLA) was performed using the Duolink Kit (Sigma-Aldrich). HEK293T cells expressing different combinations of NMDAR and BK channels were fixed with 4% paraformaldehyde for 20 min, permeabilized, and then blocked for 1 h at 37°C to avoid nonspecific antibody binding. The BK channel was detected using a rabbit polyclonal anti-MaxiK channel α subunit primary antibody (1:200, no. ab219072; Abcam). GluN1, GluN2A, and GluN2B subunits of NMDAR were detected using goat polyclonal primary antibodies anti-NMDAR1 (1:200, ref. NB100-41105; Novus Biologicals), mouse monoclonal anti-NMDAe1 (1:200, ref. sc-515148; Santa Cruz Biotechnology), and anti-NMDAe2 (1:200, ref. sc-365597; Santa Cruz Biotechnology), respectively. Secondary antibodies conjugated with oligonucleotides were supplied with the PLA DuoLink Kit. Controls consisted of non-transfected HEK293T cells or cells expressing individually the BK α subunit or single NMDAR subunits. Images were acquired on a Leica SP8 inverted confocal microscope, and image analysis was performed using the Duolink Image Tool Sigma-Aldrich) and Fiji software (Schindelin et al., 2012). The PLA technique allows the detection of protein–protein interactions (<40 nm) as quantifiable fluorescent dots (Gómez et al., 2021). About 120 cells were chosen randomly in 10 different fields from four independent experiments. Quantification was performed at the single-cell level: nuclei were automatically identified, cytoplasmic area is estimated for each cell, and PLA signal (red puncta) was normalized to individual cell area. This per-cell normalization accounts for variability in cell size and transfection efficiency and was applied consistently across all experimental replicates. Figures were graphed using Prism 10 (GraphPad).

### Electrophysiology
HEK293T cells were grown on 18-mm polylysine-treated glass coverslips and transfected as described above using the indicated combinations of plasmids. Macroscopic currents were recorded at room temperature (21–23°C) using the whole-cell patch-clamp technique with an Axopatch-700B patch-clamp amplifier (Molecular Devices) as described previously (Gómez et al., 2021). Recording pipettes were pulled from a 1.5-mm outside diameter × 0.86-mm inside diameter × 100-mm length borosilicate capillary tubes (#30-0057; Harvard Apparatus) using a programmable patch micropipette puller (Model P-97 Brown-Flaming, Sutter Instruments Co.). Micropipette resistance was 5–8 MΩ when filled with the internal solution (145 mM K-gluconate, 5 mM Mg-ATP, 1 mM EGTA, 0.2 mM Na-GTP, and 10 mM HEPES; pH 7.4) and immersed in the extracellular solution (145 mM NaCl, 5 mM HEPES, 10 mM glucose, 5 mM KCl, 2 mM $CaCl_2$, and 10 µM glycine; pH 7.4) (Gómez et al., 2021). Electrophysiological recordings were obtained using the setup described above and Clampex software (pClamp suite, Molecular Devices) at a 10,000-Hz acquisition rate and 5-kHz low-pass filter.

### Intracellular $Ca^{2+}$ fluorescence recordings
Cells were imaged using a NIKON Eclipse Ti-U microscope equipped with a Lumencor Spectra X LED, featuring a green 540-nm LED line, a 40× dry objective with a numerical aperture of 0.65, an ET-mCherry, Texas Red (Chroma) filter cube, and an iXon Ultra 888 EM-CCD camera (Andor). Fluorescent cells were

patched and recorded as described above. Micro-Manager Open Source Microscopy Software was used for fluorescence data acquisition (Edelstein et al., 2014). Fluorescent cell images were captured in 16-bit format at 4-Hz frequency acquisition. Exposure time was 100 ms. The recordings were synchronized with the amplifier via remote control using Digidata TTL-Outputs (Transistor–Transistor Logic), enabling simultaneous recording of current and fluorescence. Electrophysiology data were analyzed using pCLAMP 11 software (Molecular Devices), while fluorescence data were processed with ImageJ (Schneider et al., 2012). Briefly, images were background subtracted with the ImageJ "BG subtraction from ROI" plugin and the "Time Series Analyzer V3," plugin was used to obtain the fluorescence intensity over time. The changes in fluorescence intensity compared with the baseline fluorescence levels before the application of glutamate (delta F/F0) were graphed against time.

### Cell lysis, protein purification, and concentration determination
Total protein extracts were obtained from transfected HEK293T cells resuspended in 50 µl of lysis buffer (20 mM Tris, pH 8.0, 100 mM NaCl, 1 mM EDTA, and 0.05% Triton X-100) supplemented with protease inhibitors (Roche). After incubating for 5 min on ice, the cell suspensions were centrifuged for 10 min at $14,000 \times g$ at 4°C. Protein concentration was determined using the bicinchoninic acid assay (Smith et al., 1985).

### Cell surface biotinylation
Biotinylation and recovery of membrane proteins were carried out essentially as described before (Alvarez de la Rosa et al., 2002). Experiments were carried out at 4°C to minimize cell detachment from the plates and stop membrane trafficking. Transfected HEK293T cells were first washed with ice-cold Dulbecco's modified Eagle's medium and then twice with PBS containing 0.1 mM $CaCl_2$ and 1.0 mM $MgCl_2$ (PBS-Mg-Ca solution). Cells were then incubated with 1.5 mg/ml EZ Link Sulfo-NHS-SS-biotin (Thermo Fischer Scientific) freshly diluted into the biotinylation buffer (10 mM triethanolamine, pH 7.5, 2 mM $CaCl_2$, and 150 mM NaCl). This incubation was performed twice for 25 min at 4°C with very gentle horizontal motion to ensure thorough mixing. Cells were then rinsed twice with PBS-Ca-Mg containing 100 mM glycine and then washed in this buffer for 20 min at 4°C to quench all unreacted biotin. Cell monolayers were then rinsed twice more with PBS-Ca-Mg, and proteins were solubilized in 1 ml of lysis buffer (1.0% Triton X-100, 150 mM NaCl, 5 mM EDTA, and 50 mM Tris, pH 7.5) on ice for 60 min. Cells were then scraped, and lysates were clarified by centrifugation at $14,000 \times g$ for 10 min at 4°C. Following this, 50–100 µl of packed streptavidin-agarose beads were added to each 900 µl of supernatant and incubated overnight at 4°C with end-over-end rotation. The beads were then washed three times with lysis buffer, twice with high-salt wash buffer (similar to the lysis buffer but containing 0.1% Triton X-100 and 500 mM NaCl), and once with no-salt wash buffer (10 mM Tris, pH 7.5). Proteins were eluted from the beads in 50–100 µl of SDS-containing sample buffer.

### SDS-PAGE and western blot
Protein samples were resolved by SDS-PAGE. Samples (1 µg/µl) were prepared by mixing protein extracts with 6× Laemmli buffer (4% SDS, 20% glycerol, 10% 2-mercaptoethanol, 0.004% bromophenol blue, and 0.125 M Tris HCl, pH 6.8). SDS-PAGE was performed in Mini-PROTEAN TGX Stain-Free Precast gels (Bio-Rad). The stain-free system allowed in situ protein photoactivation after electrophoresis for total protein load visualization, quantification, and normalization. Running buffer was prepared by dilution of a 10X stock (25 mM Tris, 192 mM glycine, and 0.1% SDS) in MilliQ $H_2O$. Electrophoresis was carried out at a constant voltage of 150 V for approximately 1 h. Proteins were transferred to a polyvinylidene difluoride membrane using a Trans-Blot Turbo Transfer Starter System for western blot analysis at 1.3 A and 25 V for 10 min. Proteins of interest were visualized and detected in the polyvinylidene difluoride membranes employing the primary antibodies mouse anti-GluN2B (1:1,000 dilution, #365597; Santa Cruz Biotechnology), mouse anti-α1 $Na^+$, $K^+$-ATPase (ATP1A1) monoclonal antibody F-2 (1:1,000 dilution, sc-514614; Santa Cruz Biotechnology), and mouse anti-tubulin monoclonal antibody AA13 (T8203; Sigma-Aldrich; 1 µg/ml), followed by a secondary anti-mouse horseradish peroxidase-conjugated antibody made in goat (1:20,000 dilution, P0447; Dako). Chemiluminescence signals were recorded in a ChemiDoc imaging system and quantified using Image Lab 6.0 (Bio-Rad).

### TIRF microscopy
TIRF microscopy (TIRFM) is an optical technique that enables the excitation of fluorophores within a very thin axial region (200 nm) close to the coverslip, known as the "optical section." TIRFM was performed in a motorized Nikon Eclipse Ti microscope equipped with a 100× immersion objective. The setup included a laser unit with a diode-pumped solid state 488 laser and a 647-nm fiber laser. GFP-tagged GluN2B was visualized using a 1% diode-pumped solid state 488 laser. Images were captured using an Orca Flash 4.0 CMOS camera. To quantify the degree of GluN2B-EGFP expression at the plasma membrane, TIRF images were background subtracted using ImageJ and normalized to their integrated density and exposure time (ranging 5–100 ms). TIRFM selectively excites fluorophores within ~100–200 nm of the coverslip, capturing fluorescence from the basal plasma membrane. This produces a more uniform signal across the cell footprint, which contrasts with the peripheral signal seen in confocal images. However, shadowing or uneven illumination may still occur due to interference patterns or irregular cell topography (Ellefsen et al., 2015).

### Direct stochastic optical reconstruction microscopy combined with TIRFM
Stochastic optical reconstruction microscopy (STORM) imaging combined with TIRFM was performed on a Nikon N-STORM superresolution system with a Nikon Eclipse Ti inverted microscope equipped with an HP Apo TIRF 100× oil NA 1.49 objective (Nikon), the Perfect Focus System (Nikon), and an ORCA-Flash4.0 V2 Digital CMOS camera C11440. Fluorescence emission was filtered with a 405/488/561/640-nm Laser Quad Band

filter cube (TRF89901; Chroma). Fluorescence excitation was limited to the basal ∼200 nm of the cell using TIRFM-based illumination. This setup ensures that STORM images primarily represent the plasma membrane region adjacent to the coverslip. Thus, fluorescent signals from GluN2B and BK channels appear as a uniform distribution across the basal surface rather than lateral membrane edges. The imaging buffer specific for STORM microscopy contained 50 mM Tris-HCl (pH 8), 10 mM NaCl, 10% (wt/vol) glucose, 100 mM β-mercaptoethylamine, 0.56 mg/ml glucose oxidase, and 34 µg/ml catalase (all reagents from Sigma-Aldrich). Reconstructed images were generated from $5 \times 10^4$ acquired frames ($2.5 \times 10^4$ per channel) using the NIS-Elements software (Nikon). We performed at least three independent transfection experiments for each protein combination shown in this study. For every experiment, we determined the location of hundreds of thousands of molecules. Lateral localization accuracy was estimated, as described previously (Kshatri et al., 2020), as 13 ± 4 nm for Alexa Fluor 647 and 16 ± 6 nm for Alexa Fluor 488. Reconstructed images were filtered to remove background. Quantitative analysis of STORM images was performed using nearest-neighbor distance (NND) and cluster analysis using in-house scripts based on the k-nearest neighbor and the density-based spatial clustering of applications with noise algorithms, respectively, similar to previously published work (Kshatri et al., 2020). The clustering properties of the samples were quantified by adjusting the density filtering to 20-, 40-, or 60-nm radius with a count of 10 molecules (Fig. S1). Clusters were classified in three categories: "only red fluorophores," "only green fluorophores" (we refer to these two types as "homoclusters," formed by just one fluorophore), and "red and green fluorophores" (referred to as "heteroclusters," composed of more than one fluorophore). Cluster distributions are represented as plots of the percentage of each cluster type normalized to all clusters (all fluorophores).

### Statistical analysis
All statistical analysis was performed using Prism 10 (GraphPad). Tests are indicated in each figure legend.

### Online supplemental material
Full data analysis generated with STORM is presented in Fig. S1.

## Results
We initially focused our investigation on a set of mutations whose locations within the molecular structure of GluN2B are represented in Fig. 1. These mutations are linked to GRIN2B neurodevelopmental disorders and reported in the literature (Lemke et al., 2014; Swanger et al., 2016). We selected mutations V618G (ClinVar VCV000162085) and V15M (ClinVar VCV000375536) for our study, located in the TMD and NTD (Fig. 1). To investigate whether these disease-linked mutations on the GluN2B subunit alter the coupling of NMDAR and BK channels, we performed whole-cell voltage-clamp recordings from HEK293T cells transiently transfected with NMDARs containing the GluN1a subunit together with WT or mutant GluN2B subunits, which were co-transfected with or without the BK channel α subunit (Fig. 2).

Cells expressing GluN1-GluN2B$^{V15M}$ or GluN1-GluN2B$^{V618G}$ NMDARs produced inward currents after the application of 1 mM glutamate, very similar to those produced by GluN1-GluN2B$^{WT}$ at all potentials studied (Fig. 2 A). The characteristics of GluN1-GluN2B$^{V618G}$ were comparable with those reported for this receptor in equivalent experimental conditions (Fedele et al., 2018), whereas to our knowledge the GluN1-GluN2B$^{V15M}$ recordings shown here are the first reported to date.

Voltage-clamp recordings from cells co-expressing GluN1-GluN2B$^{WT}$ receptors with BK channels showed inward currents followed by a slower outward current at holding potentials more positive than –40 mV (Fig. 2 B), with a clear dependence on membrane voltage, similar to previously reported NMDAR-activated BK currents (Isaacson and Murphy, 2001; Zhang et al., 2018; Gómez et al., 2021). Interestingly, co-expression of GluN1-GluN2B$^{V15M}$ with BK produced comparable results (Fig. 2 B, middle panel). However, GluN1-GluN2B$^{V618G}$ failed to activate BK channels as efficiently as GluN1-GluN2B$^{WT}$ or GluN1-GluN2B$^{V15M}$ (Fig. 2 B, right panel). Effective coupling is reflected in a reduction of the net inward current flow produced by the activation of the outward current (Fig. 2 C, middle panel; see also Gómez et al., 2021). This reduction was similar in cells co-expressing BK and GluN1-GluN2B$^{WT}$ as well as GluN1-GluN2B$^{V15M}$ (Fig. 2 C, middle). In the case of cells co-expressing BK with GluN1-GluN2B$^{V618G}$, this reduction was significantly smaller, resulting in a lower decrease of charge transfer. We also quantified the efficacy of NMDAR-to-BK coupling by measuring the ratio between the inward charge and the outward charge, which we refer to as the "coupling ratio." As shown in Fig. 2 C (bottom), recordings from cells co-expressing BK with GluN1-GluN2B$^{WT}$ or with GluN1-GluN2B$^{V15M}$ showed comparable coupling ratios, whereas the values corresponding to cells co-expressing BK with GluN1-GluN2B$^{V618G}$ were significantly smaller. Altogether, these results indicate that mutation V618G in the GluN2B subunit produces selective uncoupling of NMDAR activity from BK when both proteins are co-expressed.

Several studies on the mutation V618G on GluN2B/GRIN2B have been reported (Lemke et al., 2014; Fedele et al., 2018; Vyklicky et al., 2018); however, the functional implications of this mutation have not been studied in the context of the NMDAR–BK functional association. Valine 618 is a critical and highly conserved residue located in the linker between the M2 and M3 transmembrane domains, both of which form part of the channel pore lining (Chou et al., 2022). The protein environment surrounding V618 is highly hydrophobic (Vyklicky et al., 2018). Basic properties of this mutation have been well characterized in previous studies. These have shown that this mutation does not alter the receptor's response to glutamate or glycine but is associated with decreased NMDAR open probability and lower single-channel amplitude (Vyklicky et al., 2018), contrasted by a reduction in desensitization rates and Mg$^{2+}$ block (Vyklicky et al., 2018). The latter has been related to a role of residue V618 in Mg$^{2+}$ coordination (Fedele et al., 2018). In addition, molecular dynamics studies have proposed that the V618G mutation produces a significant reorientation of the backbone carbonyl groups within the ion filter, with a significant alteration of the hydrophobicity profile (Vyklicky et al., 2018).

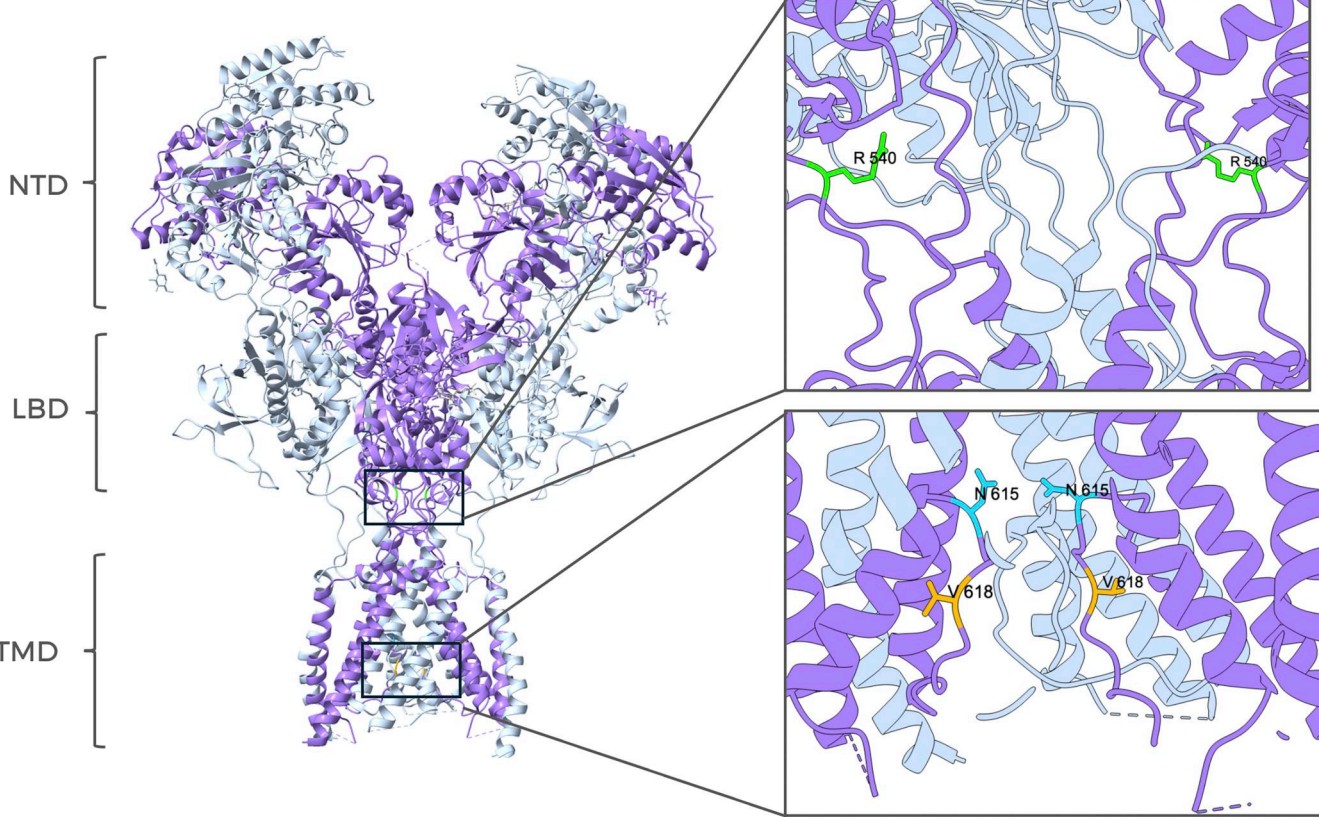

Figure 1. **Site location of disease-linked GRIN2B mutations within the structure of NMDAR (PDB:** 7SAA**) (**Chou et al., 2022**).** The GluN1 subunit is colored in gray, and the GluN2B subunit is depicted in purple. All subunits contain three main regions, indicated on the left side of the figure: N-terminal domain (NTD), ligand-binding domain (LBD), and transmembrane domain (TMD). The position of residue V15 is not shown since it is in the signal peptide (see also [Hu et al., 2016]). Mutation V618G is in the pore lining region within the TMD. The inset shows amplified images of specific regions containing disease-linked mutations. For reference, two other mutations targeting the LBD (R540H) or the TMD (N615I) are shown. The protein structure was visualized using UCSF ChimeraX (Meng et al., 2023).

Altogether, these observations suggest that the mutation would possibly result in lower selectivity and efficiency of ion permeation. This prediction seems, however, contradicted by two-electrode voltage-clamp experiments in *Xenopus* oocytes expressing GluN1-GluN2B$^{V618G}$ receptors, which showed increased Ca$^{2+}$ permeability in Mg$^{2+}$-free, NMDG-Cl solutions (Lemke et al., 2014). In contrast, a more recent study shows equivalent levels of Ca$^{2+}$ permeation of GluN1-GluN2B$^{V618G}$ compared with WT NMDARs (Fedele et al., 2018).

Considering the antecedents mentioned above, we reasoned that an alteration in Ca$^{2+}$ permeability could explain the disruption of NMDAR–BK coupling in the macrocomplexes containing GluN1-GluN2B$^{V618G}$ receptors, since the lower availability of Ca$^{2+}$ may activate fewer BK channels in the nanodomain. To assess this possibility, we co-expressed the different NMDAR–BK combinations with Lyn-R-GECO1, a genetically encoded low-affinity red fluorescent Ca$^{2+}$ indicator for optical imaging fused to a myristoylation signal peptide that targets it to the plasma membrane (Kim et al., 2016). When co-expressed with BK and GluN1-GluN2B$^{WT}$, Lyn-R-GECO1 reported the highest fluorescence, and thus Ca$^{2+}$ permeation, at the most negative potentials recorded (−60 mV), which is consistent with the driving force for Ca$^{2+}$ in our experimental design (0 mM Ca$^{2+}$ intracellular solution versus

2 mM CaCl$_2$ in the extracellular solution). Interestingly, both GluN1-GluN2B$^{V15M}$ and GluN1-GluN2B$^{V618G}$ allowed the entrance of Ca$^{2+}$ to similar extents as GluN1-GluN2B$^{WT}$, as reported by the Lyn-R-GECO1 fluorescence recordings (Fig. 3). This finding, which aligns with previous findings from Fedele et al. (2018), further suggests that the defective coupling of GluN1-GluN2B$^{V618G}$ with BK channels does not appear to be due to alterations in Ca$^{2+}$ permeability of the mutant receptors. Based on additional results reporting altered Mg$^{2+}$ permeability in GluN1-GluN2B$^{V618G}$ (Fedele et al., 2018), we also confirmed that the 5 mM Mg-ATP included in our solutions was not affecting our data. The coupling ratio obtained in symmetrical Mg$^{2+}$ solutions (5 mM MgCl$_2$) was comparable with our initial measurements (Fig. 4), eliminating the potential impact of abnormal Mg$^{2+}$ permeability on our findings.

Since Ca$^{2+}$ permeability remained unaltered in NMDAR mutants, we reasoned that differences in protein abundance and/or membrane expression of NMDAR subunits could account for the impaired coupling between GluN1-GluN2B$^{V618G}$ and BK channels. Thus, we first decided to assess overall expression levels semiquantitatively by western blot, using biotinylated membrane fractions to study protein membrane abundance (Fig. 5). The analysis of relative protein abundance in the

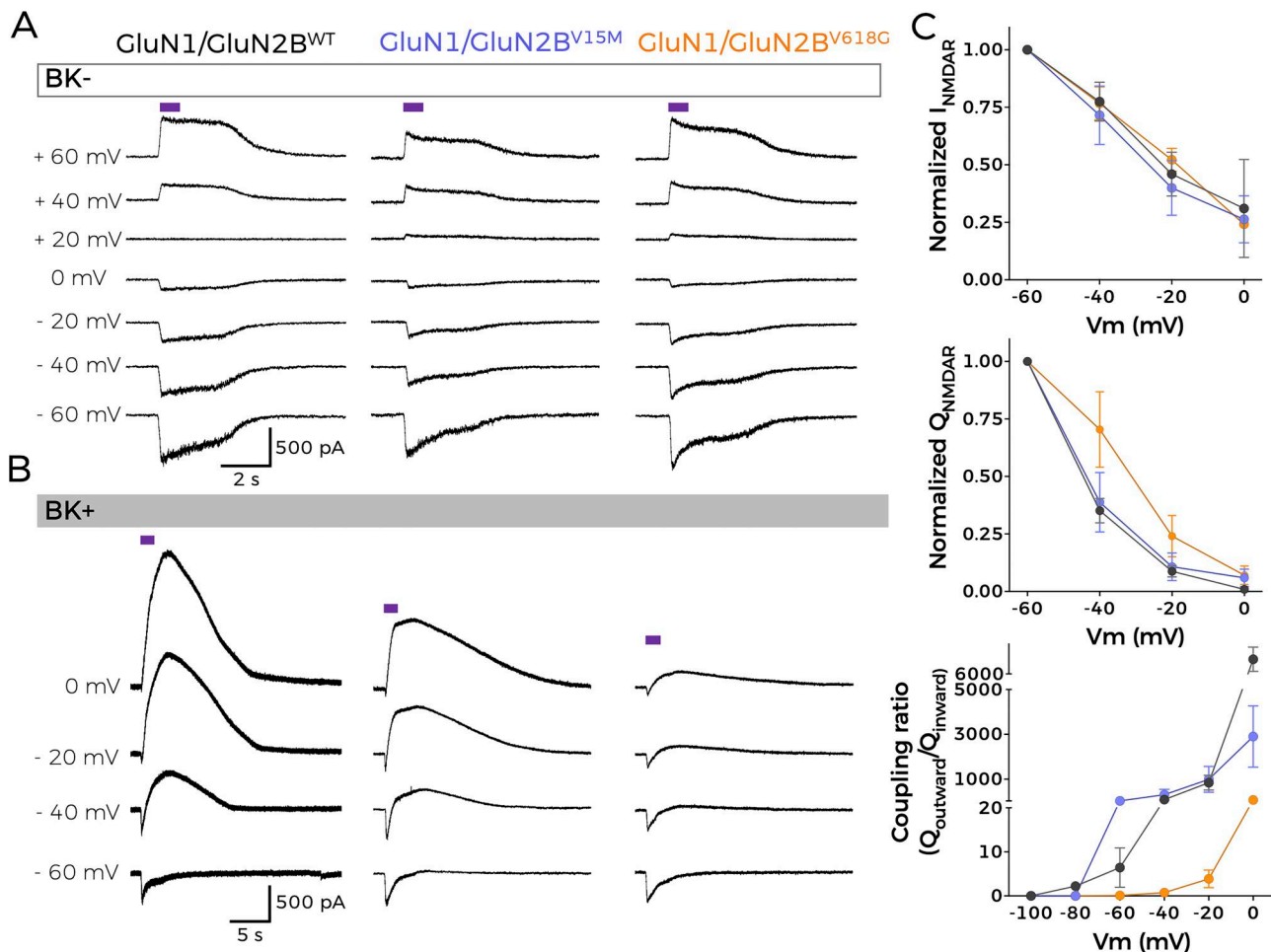

Figure 2. **Mutation V618G selectively disrupts functional NMDAR–BK coupling. (A and B)** Representative whole-cell current traces recorded from cells expressing NMDAR combinations GluN1-GluN2B$^{WT}$, GluN1-GluN2B$^{V15M}$, and GluN1-GluN2B$^{V618G}$ alone; and (B) co-expressed with BK after 1-s application of 1 mM glutamate (purple square over the traces) evoked at the indicated membrane potentials. **(C)** Normalized I-V (top) and charge-voltage (Q-V, middle) relationships for NMDAR inward currents. All plots refer to NMDAR/BK currents. Bottom: $Q_{outward}/Q_{inward}$ relationships versus voltage. Data points in all graphs represent mean ± SEM; $n$ = 5–7.

cell revealed an increase in the expression levels of GluN1-GluN2B$^{V618G}$ in comparison with GluN1-GluN2B$^{WT}$ (Fig. 5, A and B). In contrast, the variant V15M showed diminished expression levels. Notably, these experiments also pointed toward the increased membrane abundance of GluN1-GluN2B$^{V618G}$ with respect to GluN1-GluN2B$^{WT}$ (Fig. 5 C). It is important to note that we did not observe consistent differences in the viability of GluN1-GluN2B$^{V15M}$– or GluN1-GluN2B$^{V618G}$–transfected cells (by general appearance of the culture, or total protein level comparison), so we discarded this as a possible factor to explain our results.

In view of the results above, and to perform a precise quantification of the membrane abundance levels of the two mutants, we used TIRFM (Fig. 6). This technique enables the selective excitation of surface-bound fluorophores, allowing us to study quantitatively the membrane population of NMDARs containing the different GluN2B variants. HEK293T cells were transfected with the different NMDAR combinations using GFP-tagged GluN2B subunits. As shown in Fig. 6 B, the GluN1-GluN2B$^{V618G}$ NMDARs showed significantly increased expression at the plasma membrane, whereas GluN1-GluN2B$^{V15M}$ membrane abundance was significantly lower than GluN1-GluN2B$^{WT}$, in agreement with the biotinylation data.

We then tested whether BK differentially associates with specific NMDAR subunit variants by using PLA as previously reported (Gómez et al., 2021). This technique is based on the combination of antibody-based protein recognition and nucleotide-based rolling circle amplification, enabling the detection of protein proximity within a radius of 40 nm (Alam, 2018; Gómez et al., 2021). Consistent with our previous findings, positive PLA signals were observed for HEK293T cells co-expressing BK and GluN1-GluN2B$^{WT}$, demonstrating that BK channels and NMDARs formed nanodomains in our experimental conditions (Fig. 7 A). Cells transfected with GluN1-GluN2B$^{V15M}$ in the presence of BK channels show a marked reduction in the number of positive PLA signals (Fig. 7). This result matched the lower membrane abundance of this variant. Most strikingly, PLA signals for cells transfected with GluN1-GluN2B$^{V618G}$ and BK channels were also diminished in comparison with BK and GluN1-GluN2B$^{WT}$.

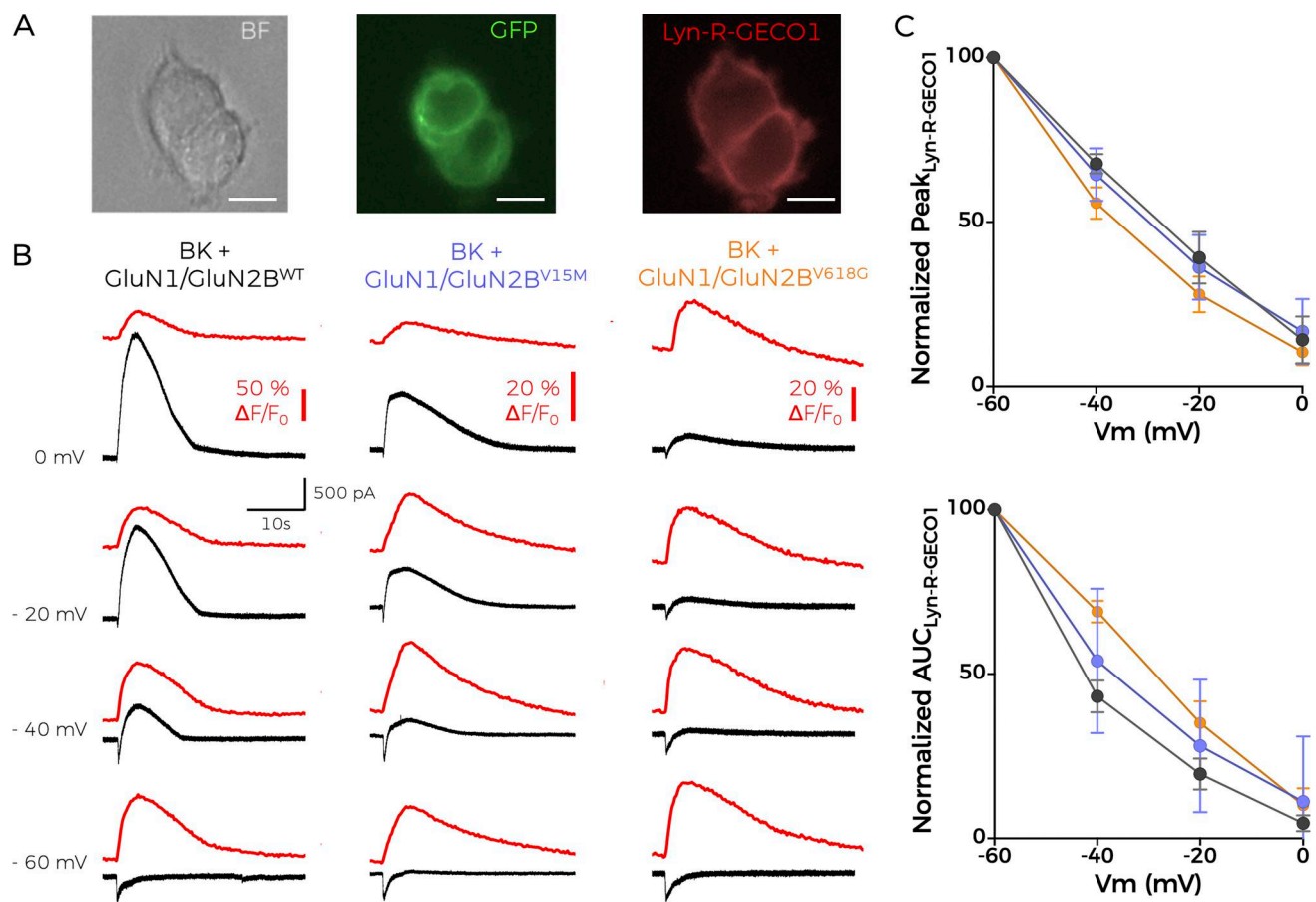

Figure 3. **GluN2B disease-linked mutants are permeable to Ca²⁺.** **(A)** Representative microscopy images showing HEK cells co-transfected with BK plus NMDARs containing the GFP-tagged GluN2B mutants used in this study, as well as the red fluorescent, membrane-linked Lyn-R-GECO1 Ca²⁺ indicator. Images were obtained in bright field (BF, left panel), with 488-nm excitation light (middle) and with 540-nm excitation light (right panel). Scale bar = 10 μM. **(B)** Simultaneous whole-cell current (black traces) and normalized fluorescence recordings ($\Delta F/F_0$, red traces) from cells co-expressing BK channels, the membrane calcium sensor Lyn-R-GECO1 and GluN1-GluN2B^WT (left), GluN1-GluN2B^V15M (middle), or GluN1-GluN2B^V618G (right). Recordings were obtained at the indicated holding potentials after application of 1 mM glutamate for 1 s. **(C)** Graphs represent the averaged maximal peak from Lyn-R-GECO1 recordings (top) or the normalized area under the curve (AUC) (bottom) versus voltage relationships corresponding to the experiments shown in B, in the absence of BK; Data points represent mean ± SEM; $n$ = 5–7.

We did not anticipate this result, given the large increase in membrane abundance of GluN1-GluN2B^V618G in comparison with GluN1-GluN2B^WT (Figs. 5 and 6).

Altogether, our data show that the functional coupling of GluN1-GluN2B^V618G receptors to BK channels is significantly diminished as compared with that of GluN1-GluN2B^WT or another disease-linked mutant, GluN1-GluN2B^V15M. This effect occurs despite a higher membrane abundance of GluN1-GluN2B^V618G but is consistent with the observed reduction of protein–protein interactions between GluN1-GluN2B^V618G and BK channels reported by PLA experiments. Two scenarios may possibly explain these results. On one hand, mutant receptors GluN1-GluN2B^V618G and BK channels could be located at further distances within the complexes. Another possibility may be the alteration of multichannel cluster characteristics (size, composition, or a combination of both) when GluN1-GluN2B^V618G and BK channels are co-expressed. To discern whether any of these possibilities, or a combination of all of them, may reconcile our observations and enlighten the cellular mechanism underlying the altered

functional coupling of GluN1–GluN2B^V618G–BK complexes, we used STORM superresolution microscopy. This technique was combined with TIRFM to investigate the spatial organization of NMDAR–BK complexes at or near the plasma membrane (Figs. 8 and 9) (Kshatri et al., 2020).

Close localizations of BK and GluN2B^WT were observed, as reflected in the NND distribution analysis of the STORM images, which showed a higher peak at 25–30 nm (Fig. 8 B). These data support previous findings indicating that BKα and GluN1-GluN2B^WT are in nanoscale proximity (Zhang et al., 2018; Gómez et al., 2021). Surprisingly, the BKα-to-GluN2B distance distribution within the nanodomains range (0–50 nm) was practically identical in cells co-expressing BK with GluN1-GluN2B^WT or with BK GluN2B^V618G (Fig. 8 C). These results indicate that the presence of the V618G mutation in the GluN2B subunit does not increase the distance between BK and NMDAR in the nanodomains.

Cluster analysis of superresolution data provides useful insights into spatial patterns and associations between proteins

Figure 4. **NMDAR$^{V618G}$–BK coupling efficiency is unaltered in symmetrical Mg$^{2+}$ solutions. (A)** Location of residues involved in the coordination of the Mg$^{2+}$ block: N616 in GluN1, and N615 and V618G in GluN2B (PDB accession no. 7SAA) (Chou et al., 2022). **(B)** Normalized I-V graphs obtained from quantification of whole-cell currents recorded from HEK293T cells co-expressing BK with GluN1/GluN2B$^{V618G}$ in the absence (red symbols) or presence of symmetrical Mg$^{2+}$ (blue symbols). **(C)** Normalized charge-voltage (Q-V) relationships for NMDAR inward currents from the experiments described in B. **(D)** Functional coupling between BK and NMDAR estimated as $Q_{outward}/Q_{inward}$ relationships versus voltage.

(Ricci et al., 2015; Vivas et al., 2017; Zanacchi et al., 2017; Kshatri et al., 2020). We performed this analysis to better understand whether there are any differences in cluster formation between BK/GluN1-GluN2B$^{WT}$ and BK/GluN1-GluN2B$^{V618G}$. We used in-house software written in Python to identify and calculate areas of clusters with all possible protein combinations in each experimental condition. This analysis applies the density-based spatial clustering of applications with noise algorithm, a data-clustering algorithm that finds core samples of high density and expands clusters from them. This algorithm is based on two parameters set by the experimenter referring to the radius of the core cluster and to the minimum number of particles contained in it (Ricci et al., 2015; Kshatri et al., 2020). Based on previous experience, we analyzed clusters, setting the core size to 10 particles (Kshatri et al., 2020), and generated three full analyses considering radii of 60, 40, and 20 nm (Fig. S1). A striking result was that, in all conditions tested, the size of BK/GluN1-

GluN2B$^{WT}$ heteroclusters was significantly larger than that of heteroclusters formed by BK and GluN1-GluN2B$^{V618G}$, as inferred from the analysis of the cumulative probability of cluster area (Fig. 9 A and Fig. S1) (Vivas et al., 2017). In addition, comparison of all calculated distributions consistently showed the following (Fig. 9 B and Fig. S1): (1) The proportion of NMDAR–BK heteroclusters in cells expressing BK and GluN1-GluN2B$^{WT}$ was very similar to that observed in cells expressing BK and GluN1-GluN2B$^{V618G}$; (2) BK homoclusters are more abundant in cells co-expressing BK with GluN1-GluN2B$^{WT}$ than in those co-expressing BK with the disease-linked mutant GluN1-GluN2B$^{V618G}$; and (3) NMDAR homoclusters are more abundant in cells co-expressing BK with GluN1-GluN2B$^{V618G}$ than in those co-expressing BK with GluN1-GluN2B$^{WT}$. The latter observation is consistent with the biotinylation and TIRFM data shown in this study (Figs. 5 and 6).

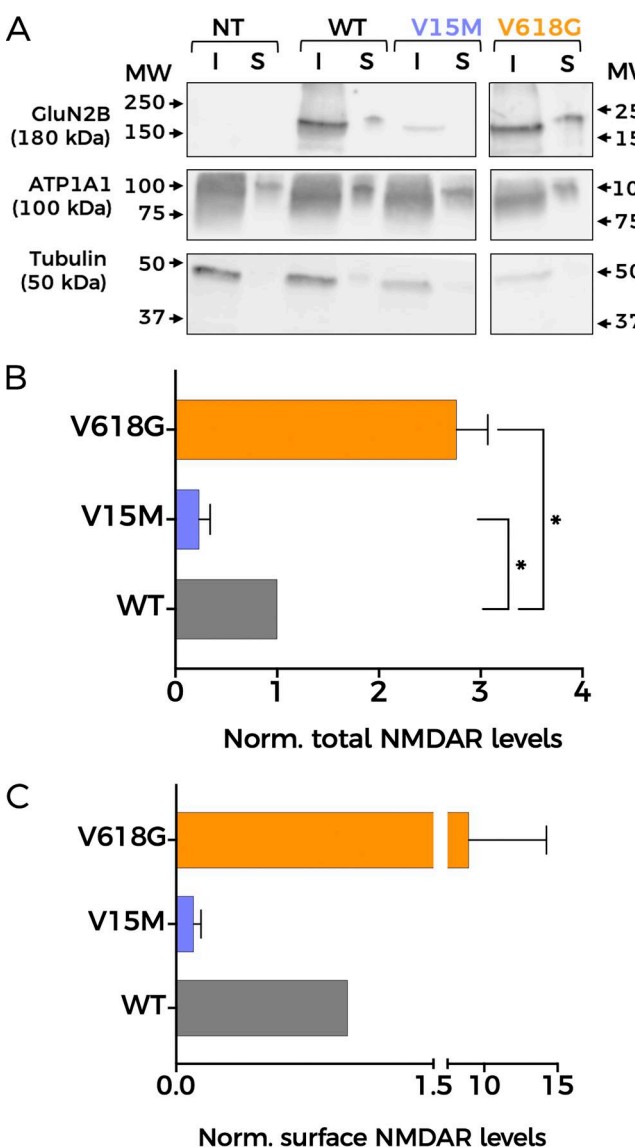

Figure 5. **Mutation V618G shows increased membrane abundance.** **(A)** Representative western blot of NMDAR protein abundance in HEK293T cells non-transfected (NT) or transfected with GluN1-GluN2B[WT] (WT), GluN1-GluN2B[V15M] (V15M), or GluN1-GluN2B[V618G] (V618G). Input (I): total lysate; surface (S): biotinylated fractions corresponding to membrane proteins. Middle panel: Na⁺/K⁺ ATPase α1 subunit (ATP1A1). Lower panel: tubulin. All sections of the image correspond to the same experiment. Some lanes have been omitted since they are not relevant for this study. Arrows indicate migration of molecular mass markers (MW; values in KDa). **(B)** Protein abundance for the indicated GluN2B subunit variants calculated as the ratio GluN2B/tubulin levels, then normalized to WT values. $N = 4$; *, $P < 0.05$ (one-way ANOVA/Dunnett's test). **(C)** Total GluN2B membrane abundance calculated as the ratio GluN2B/ATP1A1 membrane fractions (S), normalized to WT membrane values. All graphs: data points represent mean ± SEM. Source data are available for this figure: SourceData F5.

## Discussion

In this work, we provide additional evidence that BK and NMDAR nanodomains can be functionally reconstituted in a heterologous expression system such as HEK293T, offering a valuable model system to understand the mechanisms underlying the formation and function of these channelosomes.

Functional coupling of GluN1-GluN2B[WT] to BK was recorded electrophysiologically and recapitulated the biophysical properties previously described in neurons (Isaacson and Murphy, 2001; Zhang et al., 2018; Gómez et al., 2021). PLA and super-resolution analysis showed that BK and GluN1-GluN2B[WT] are located in nanoscale proximity, showing a sharp NND maximal peak at around 25 nm. The nanoscale proximity of BK channels and NMDARs is a critical aspect of their functional relationship in neurons, facilitating efficient Ca²⁺ signaling and modulating neuronal excitability, synaptic transmission, and plasticity (Gómez et al., 2021).

We discovered that two disease-linked mutations in the GluN2B subunit, V15M and V618G, alter formation of BK–NMDAR complexes. A striking observation of this study is that NMDARs containing GluN2B[V618G] subunits showed disrupted functional coupling to BK channel function, as demonstrated by measuring the coupling ratio from whole-cell current recordings, in spite of significantly higher membrane expression and normal Ca²⁺ permeability compared with GluN1-GluN2B[WT]. In contrast, GluN1-GluN2B[V15M] showed functional coupling comparable with GluN1-GluN2B[WT] regardless of its significantly lower membrane abundance. This reduction is most likely attributable to the position of the V15 residue within the signal peptide at the extreme N terminus, which directs the nascent GluN2B protein to the plasma membrane. In both rat and human GluN2B sequences (RefSeq NP_036706.1 and NP_001400921.1, respectively), the signal peptide comprises the first 26 amino acids and is cleaved during protein maturation, which explains its absence from resolved structural models. Beyond GluN2B, a genome-wide analysis of pathogenic signal peptide variants has shown that such mutations can impair protein targeting, translocation, processing, and stability (Gutierrez Guarnizo et al., 2023), consistent with our findings. Although other GRIN2B pathogenic variants within the signal peptide—such as V18I—have been reported (Hu et al., 2016), to our knowledge, this study presents the first detailed functional characterization of the V15M mutation. Our results suggest that its likely pathogenic effect stems from reduced protein expression and impaired membrane localization. This could in turn lead to a decreased formation of NMDAR–BK nanodomains in neurons, although this remains to be explored.

Mutation V618G, however, poses an interesting conundrum. How can a mutation inside the NMDAR pore contribute to the disruption of the functional coupling between GluN1-GluN2B[V618G] and BK channels? Some pore mutations can destabilize channel openings by altering the receptor conduction pathway (Tristani-Firouzi et al., 2002), while other pore mutants may reshape the pore cavity and alter the channel's selectivity filter (Cordero-Morales et al., 2006). This might have been the case for mutant V618G, with some studies reporting altered Ca²⁺ and Mg²⁺ permeability, (Lemke et al., 2014; Vyklicky et al., 2018). However, and in agreement with previous reports showing comparable Ca²⁺ permeation properties between GluN1-GluN2B[V618G] and GluN1-GluN2B[WT] (Fedele et al., 2018), our results demonstrated that the disruption in NMDAR–BK functional coupling of GluN1-GluN2B[V618G] could not be ascribed to differences in the permeation of Ca²⁺, as shown with simultaneous

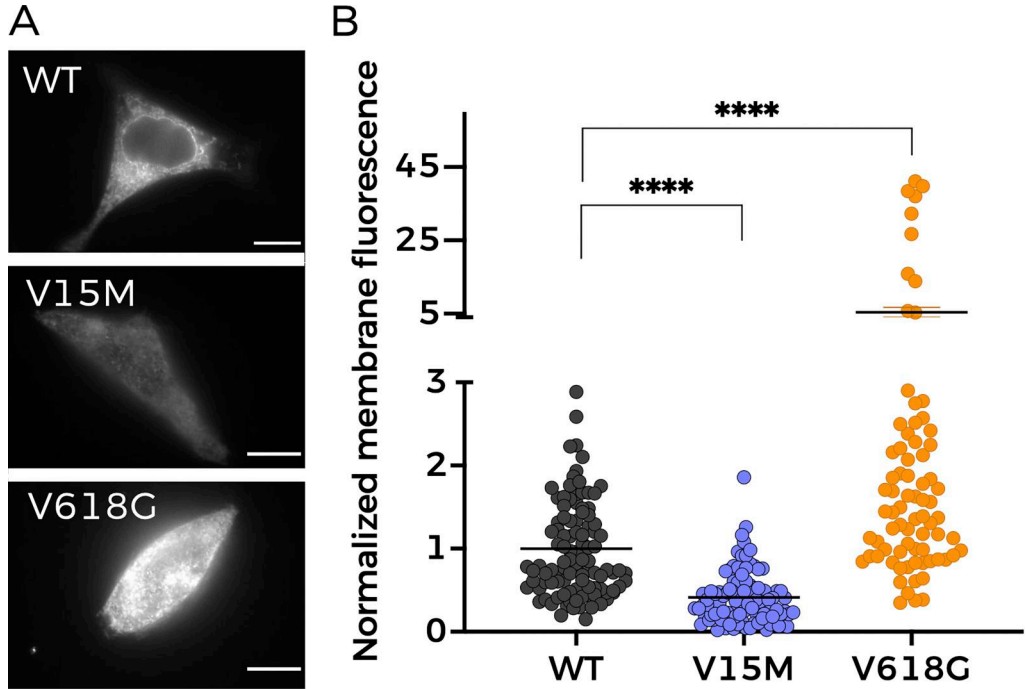

**Figure 6. Membrane abundance of different NMDAR combinations measured with TIRFM. (A)** Representative TIRFM images of HEK293T cells transiently transfected with NMDAR containing combinations of GluN1a and either WT- or mutant-GluN2B subunits. Scale bar = 5 µm. **(B)** Quantitative analysis of TIRFM imaging from cells expressing heteromers of GluN1a and the indicated GluN2B variants, normalized to the fluorescence levels of GluN1-GluN2B[WT] (black circles). Data represent mean ± SEM (minimum $n$ = 30 cell counts per condition and experiment; three independent experiments). ****, P < 0.0001 (Kruskal–Wallis/Dunn's test).

$Ca^{2+}$ and voltage-clamp recordings. Additionally, we demonstrated that, even if the V618G mutant showed altered $Mg^{2+}$ permeation or block, this could not account for the altered coupling of BK and NMDAR in the nanodomains. It could be argued that, while the experimental conditions used in our

experiments allow to compare the coupling to BK of NMDAR[WT] and the mutants in the absence of $Mg^{2+}$, they may underscore the extent to which NMDAR[V618G] and BK are coupled in physiological conditions. Fedele et al. (2018) showed that $Mg^{2+}$ block is lost in the V618G mutant, meaning that these receptors should

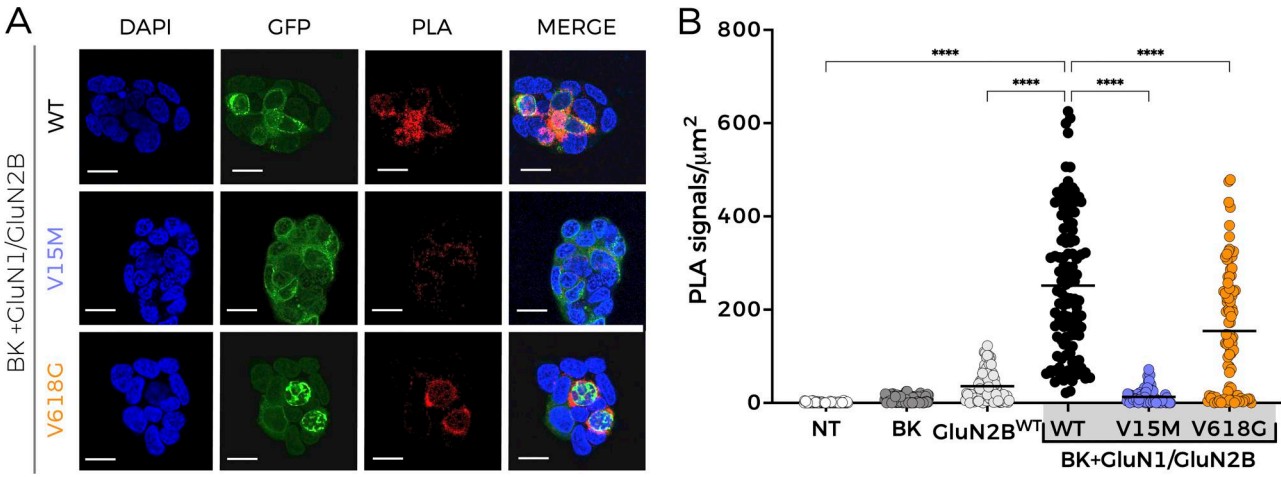

**Figure 7. Reduced protein–protein interactions between BK and NMDAR containing disease-linked GluN2B subunits revealed by PLA. (A)** Representative confocal microscopy images of PLA experiments in HEK293T cells co-expressing BK with NMDAR, including the GluN2B variants indicated on the left. Each column corresponds to an imaging channel (left, DAPI, 405 nm; middle, NMDAR-GFP, 488 nm; right, PLA, 540 nm); merged channels are shown at the far-right column. Scale bar is 20 µm. **(B)** Quantification of PLA signals normalized to cell area (µm²) for HEK293T cells non-transfected (NT) or transiently expressing the protein combinations indicated in the axis. Data points represent individual cells; horizontal bars correspond to mean values. $n$ = 35–45 cell counts per experiment; four independent experiments. ****P < 0.0001 (Kruskal–Wallis/Dunn's test).

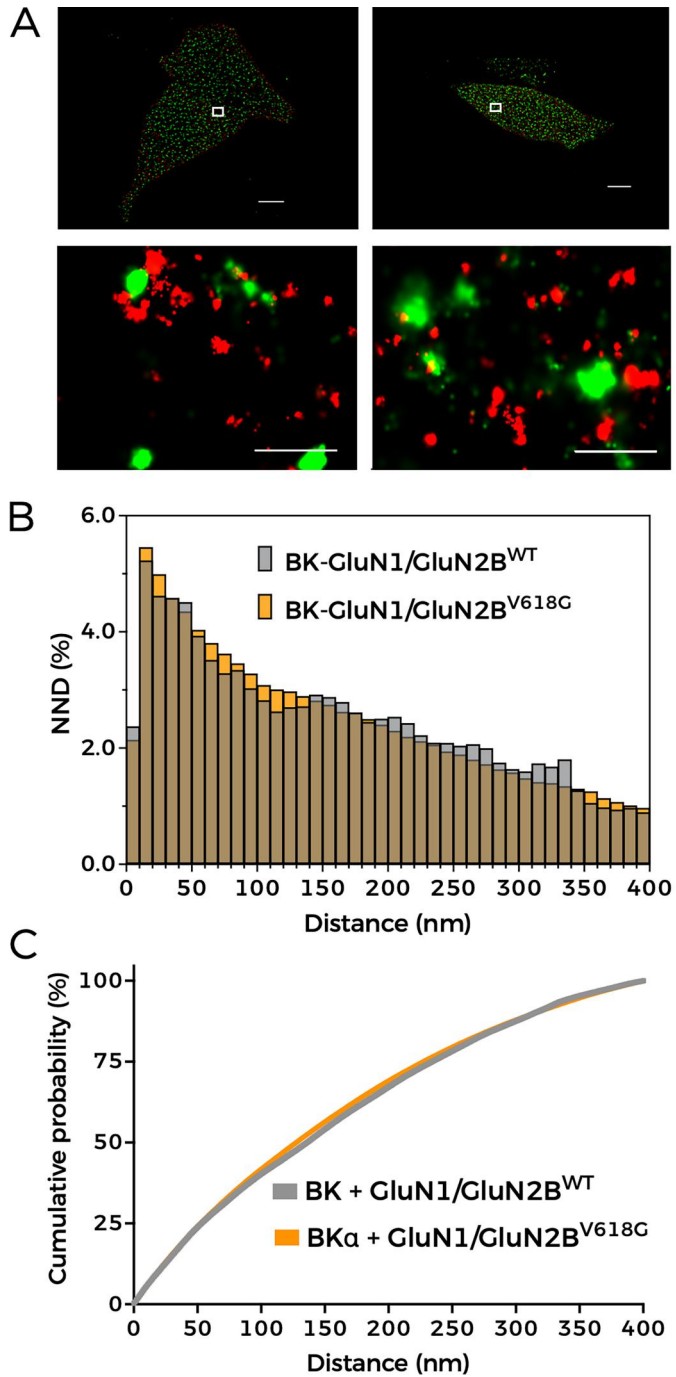

Figure 8. **BK and NMDAR are in nanoscale proximity. (A)** Representative STORM images (top; scale bar = 5 μM) and magnified views of areas of interest (bottom; scale bar = 0.5 μM) showing the spatial distribution of BKα (green, Alexa Fluor 488) and GluN2B (red, Alexa Fluor 647) in HEK293T cells co-expressing BK, GluN1a, and either GluN2B$^{WT}$ (left) or GluN2B$^{V618G}$ (right). **(B)** NND analysis from the corresponding dual-label experiments indicated in the graph legend. **(C)** Cumulative probability analysis of NND distribution.

be activated to a much greater extent than WT channels at resting membrane potentials. However, we reason that, even though increased Ca$^{2+}$ would permeate through NMDAR$^{V618G}$ at more negative potentials, this may not result in physiologically relevant NMDAR–BK coupling. This is basically due to the fact that BK is not only activated by Ca$^{2+}$ but also by voltage. Our previous data allowed us to estimate the shift in the BK G–V

curve produced by Ca$^{2+}$ entering through nearby NMDAR in nanodomains from excised membrane patches (Fig. 3 from Gómez et al. [2021]). In physiological conditions and with NMDAR$^{WT}$, the position of this voltage-activation curve, which depends on the Ca$^{2+}$ concentration surrounding BK, predicts that, at resting membrane potentials (around –70 mV), most BK channels will be closed. Even if the Ca$^{2+}$ concentration is largely

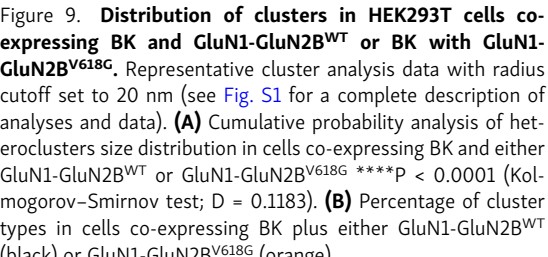

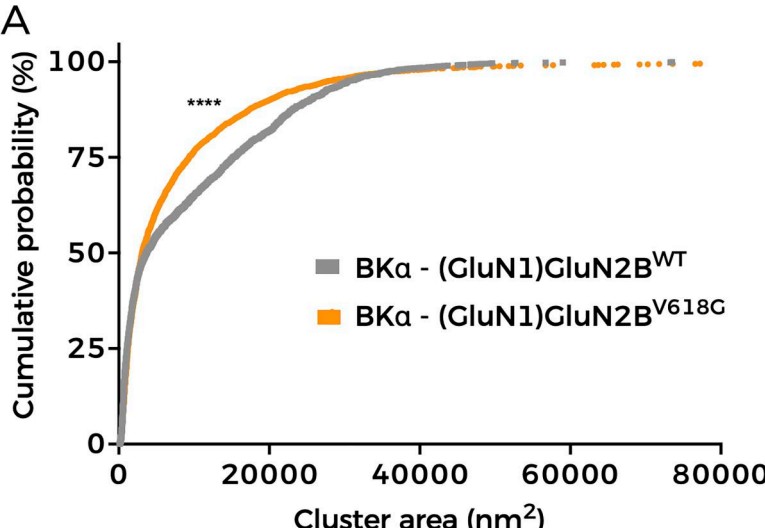

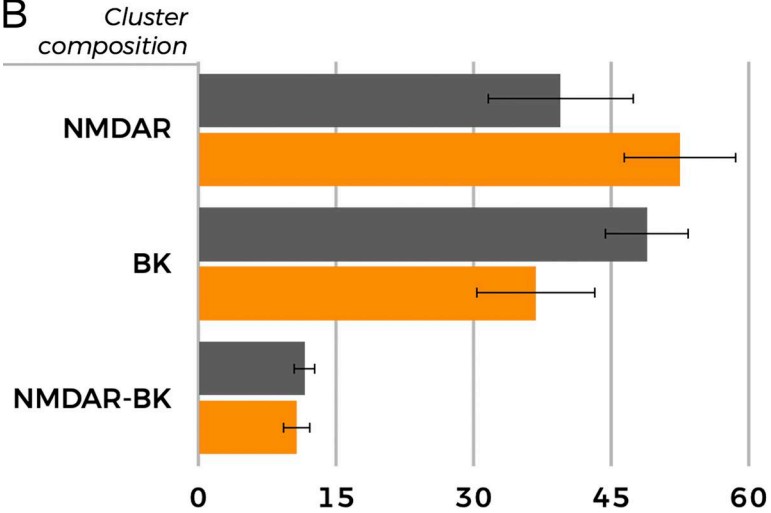

Figure 9. **Distribution of clusters in HEK293T cells co-expressing BK and GluN1-GluN2B^WT or BK with GluN1-GluN2B^V618G.** Representative cluster analysis data with radius cutoff set to 20 nm (see Fig. S1 for a complete description of analyses and data). **(A)** Cumulative probability analysis of heteroclusters size distribution in cells co-expressing BK and either GluN1-GluN2B^WT or GluN1-GluN2B^V618G ****P < 0.0001 (Kolmogorov–Smirnov test; D = 0.1183). **(B)** Percentage of cluster types in cells co-expressing BK plus either GluN1-GluN2B^WT (black) or GluN1-GluN2B^V618G (orange).

increased, as expected in the case of the NMDAR$^{V618G}$ mutant, the amount of available BK channels at negative voltages would be significantly low. Therefore, the increased Ca$^{2+}$ permeability at negative voltages would not result in effective coupling, at least not to the extent of that observed at positive membrane voltages with WT channels. It is important to note that while our biotinylation and TIRFM experiments (Figs. 5 and 6) revealed significant differences in membrane abundance among GluN2B variants, these differences were not mirrored in the raw NMDAR current amplitudes presented in Fig. 2 A. This apparent discrepancy likely stems from the fact that our study focused primarily on assessing NMDAR–BK coupling efficiency, rather than conducting a comprehensive biophysical comparison of the isolated NMDAR variants. Representative traces in Fig. 2 B were selected to ensure comparable inward NMDAR currents, allowing for a clearer evaluation of coupling differences. Additionally, the coupling efficiency metric (Q_outward/Q_inward) is inherently relative, and thus reflects the functional outcome of both expression and interaction within each specific condition. Furthermore, current densities were not normalized to cell size (pA/pF) and therefore do not provide a quantitative assessment of

surface receptor abundance. The fluorescence-based Ca$^{2+}$ imaging in Fig. 3, presented as normalized $\Delta F/F_O$ values, similarly served to illustrate that all three NMDAR variants support Ca$^{2+}$ influx, rather than quantify absolute permeation levels. Together, these design choices were guided by the central aim of the study—to evaluate how GRIN2B mutations affect NMDAR–BK nanodomain coupling—rather than to recharacterize the well-studied properties of the V618G variant or fully define those of V15M. Altogether, our results support the idea that the mechanism underlying the disrupted NMDAR–BK coupling associated to mutation V618G is due to a defect of the cell biology of the complex formation. The fact that NMDAR–BK complexes do not form correctly even in the presence of enhanced plasma membrane expression of GluN1-GluN2B$^{V618G}$ reinforces this hypothesis. Even though PLA experiments showed a diminished number of protein–protein interactions, the subsequent analysis using superresolution microscopy discarded a potential increase in the distance between NMDAR and BK in the nanoclusters, since the distance distribution between both proteins remained unaltered. In fact, the combination of PLA and superresolution microscopy demonstrated that complex formation still occurs, albeit with

altered size and distribution of the nanodomains. At this point, it is important to note an apparent contradiction between the experimental results: while the PLA data show a reduced number of interactions, STORM data indicate that the proportion of NMDAR$^{WT}$–BK and NMDAR$^{V618G}$–BK heteroclusters is similar. In fact, both experimental results are only reconcilable if the nanocluster size is reduced in the case of the mutant, since the smaller size of the mutant clusters would reduce the probability of antibody interaction within the NMDAR–BK complex, which explains the lower number of interactions shown by the PLA data.

Interestingly, we observed a different proportion of BK and NMDAR homoclusters between WT and mutant conditions. Augmented expression and plasma abundance of GluN1-GluN2B$^{V618G}$, demonstrated by western blot, biotinylation experiments and TIRF, should be reflected in an increase of the proportion of homoclusters, as detected in our STORM experiments. However, if the constitution of the nanodomains followed similar mechanisms as with NMDAR$^{WT}$, we would also expect a reflection of the increased NMDAR$^{V618G}$ levels on a larger proportion of NMDAR–BK heteroclusters. The fact that this is not observed indicates that heterocomplexes formation is impaired in the case of the V618 mutants or occurs with lower efficiency. These observations lead us to conclude that, within the heterocomplexes, the proportion of GluN1-GluN2B and BK particles must be different in nanodomains containing BK/GluN1-GluN2B$^{V618G}$ and those containing BK/GluN1-GluN2B$^{WT}$. Another factor to consider would be the possibility that co-expression of the NMDA variants is affecting BK expression levels. STORM labeling did not support this idea, as the total BK fluorescent signal was comparable across all tested conditions (Fig. S1). In summary, our data is consistent with the idea that efficient NMDAR–BK functional coupling requires an adequate proportion between both channels and likely a minimum number of participating units in the nanodomain.

Currently, there is very limited information regarding molecular determinants of NMDAR–BK nanodomain formation. Zhang et al. (2018) showed that the S0–S1 loop in the α subunit of BK interacts with intracellular regions of the GluN1 subunit. Our results do not contradict this model but suggest that GluN2B subunits may also participate in regulating the interaction of NMDAR with BK channels. The question remains how a pore mutation such as V618G may alter such interactions. It is tempting to speculate that this mutation may allosterically disrupt a protein–protein interface that either interacts directly with the BKα subunit or, alternatively, induces changes in GluN1, which in turn alters the interaction with BK. Our results do not allow us to distinguish between these two possibilities. A broader implication of our results is that the presence of different GluN2 regulatory subunits may introduce diversity in the biophysical properties of the nanodomains, and thus in their physiological roles, such as the fine-tuning of synaptic plasticity (Zhang et al., 2018; Gómez et al., 2021). Clearly, a deeper understanding of the structural and dynamic properties of NMDAR–BK complex formation is warranted, both from the perspective of their physiological role and as a basis for the pathophysiological consequences of disease-causing mutations.

In summary, we have uncovered a mutation, GluN2B$^{V618G}$, that selectively alters BK–NMDAR complex formation and functional coupling, an effect that may underlie at least some of its pathogenic effects on GRIN2B-related neurodevelopmental disorder patients and suggests mechanisms by which BK–NMDAR complexes may modulate synaptic transmission and neuronal function.

## Data availability

All study data are included in the article and/or supplemental material. Additional information can be provided by the authors upon reasonable request.

## Acknowledgments

Crina M. Nimigean served as editor.

We thank Tatiana Labouré, from the Institut de Biologie Structurale, for useful discussions related to hydrophobicity analyses; Carlo Manzo, from University of Vic, for helpful comments related to the analysis of the superresolution data; and all members of the Giraldez lab for constructive discussions during the elaboration of this manuscript.

This work was supported by grant PID2021-128668OB-I00 funded by MICIU/AEI (10.13039/501100011033) and by "ERDF/EU" (to T. Giraldez). Grant PRE2019-089248 funded by MICIU/AEI (10.13039/501100011033) and "ESF Investing in your future" (to R. Martínez-Lázaro). Grant FJC2020-042989-I funded by MICIU/AEI (10.13039/501100011033) (to T. Minguez-Viñas).

Author contributions: R. Martinez-Lázaro: conceptualization, data curation, formal analysis, investigation, methodology, project administration, software, validation, visualization, and writing—original draft, review, and editing. T. Minguez-Viñas: conceptualization, formal analysis, investigation, and writing—review and editing. A. Reyes-Carrión: conceptualization, investigation, and writing—original draft, review, and editing. R. Gómez: conceptualization, methodology, and supervision. D. Alvarez de la Rosa: conceptualization, formal analysis, validation, and writing—review and editing. D. Bartolomé-Martín: conceptualization, investigation, methodology, supervision, and writing—original draft, review, and editing. T. Giraldez: conceptualization, formal analysis, funding acquisition, investigation, methodology, project administration, resources, supervision, validation, visualization, and writing—original draft, review, and editing.

Disclosures: The authors declare no competing interests exist.

Submitted: 17 March 2025

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

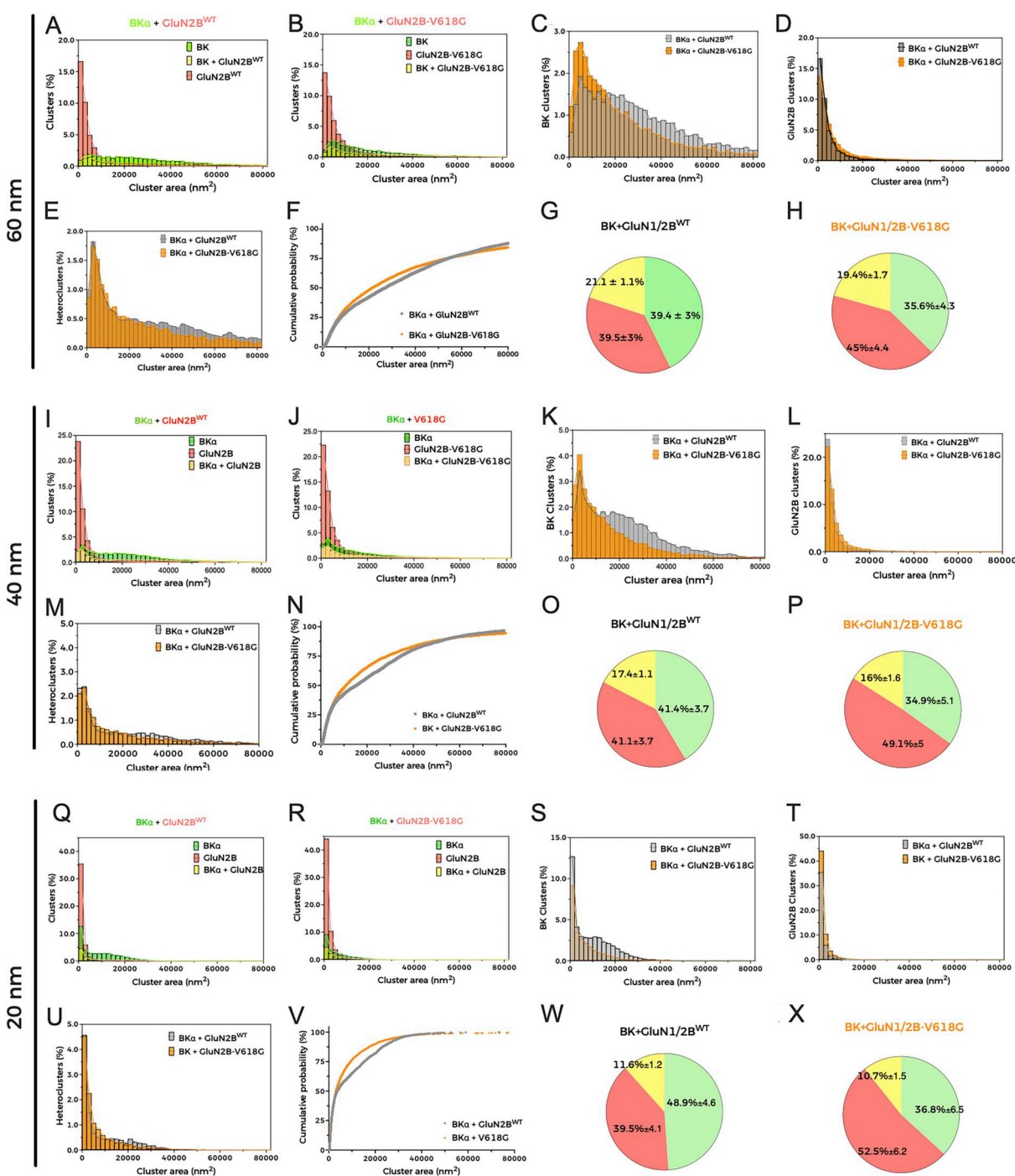

Figure S1. **Cluster analysis of BK and GluN2B.** Graphs and analysis are distributed in three groups corresponding to three different settings where the cluster radius is fixed to 60 nm (top), 40 nm (middle), or 20 nm (bottom). **(A, B, I, J, Q, and R)** Histograms representing the distribution of clusters in HEK293T cells co-expressing either BK and GluN1/2B^WT or BK and GluN2B-V618G, containing BKα alone (green bars), GluN2B^WT alone (red bars), or both proteins (yellow bars). Colored curves outline the histograms to facilitate visualization. In all cases, the clustering properties of the samples were quantified by adjusting the density filtering with a count of 10 molecules and radius cutoff set to (A and B) 60 nm (r = 60 nm), (I and J) 40 nm (r = 40 nm), and (Q and R) 20 nm (r = 20 nm). **(C, K, and S)** Distributions of BK homoclusters in HEK293T cells co-expressing either BK and GluN1/2B^WT (gray bars) or BK and GluN1/2B-V618G (orange bars) obtained with the different radius values (C, r = 60 nm; K, r = 40 nm; S, r = 20 nm). **(D, L, and T)** Distributions of GluN2B homoclusters in HEK293T cells co-expressing either BK and GluN1/2B^WT (gray bars) or BK and GluN1/2B-V618G (orange bars) obtained with the different radius values (D, r = 60 nm; L, r = 40 nm; T, r = 20 nm). **(E, M, and U)** Distributions of NMDAR–BK heteroclusters in HEK293T cells co-expressing either BK and GluN1/2B^WT (gray bars) or BK and GluN1/2B^V618G (orange bars) obtained with the different radius values (E, r = 60 nm; M, r = 40 nm; U, r = 20 nm). **(F, N, and V)** Cumulative probability analysis of heterocluster distribution in cells co-expressing BK and either GluN2B^WT or GluN2B-V618G at the different values of radius (F, r = 60 nm, [Kolmogorov–Smirnov test {K–S} ****P < 0.0001, D = 0.06232]; N, r = 40 nm, [K–S test ****P < 0.0001, D = 0.09336]; V, r = 20 nm [K–S test ****P < 0.0001, D = 0.1183]). **(G, H, O, P, W, and X)** Pie charts illustrating the proportion (%) of cluster distribution in cells co-expressing BK and either GluN1/2B^WT or GluN1/2B^V618G at radius values (G and H) r = 60 nm, (O and P) r = 40 nm, and (W and X) r = 20 nm.

