## [Peer Review File · The Journal of General Physiology]

GRIN2B disease-associated mutations disrupt the function of BK channels and NMDA receptor signalling nanodomains

Rebeca Martinez-Lazaro, Teresa Minguez-Viñas, Andrea Reyes-Carrión, Ricardo Gómez, Diego Alvarez de la Rosa, David Bartolomé-Martín, and Teresa Giraldez

Corresponding Author(s): Teresa Giraldez, University of La Laguna

Review Timeline:

Submission Date:	March 17, 2025
Editorial Decision:	April 9, 2025
Revision Received:	May 8, 2025
Editorial Decision:	May 27, 2025
Revision Received:	July 5, 2025

Editor: Crina Nimigean

Transaction Report:

DOI: <https://doi.org/10.1085/jgp.202513799>

April 10, 2025

Prof. Teresa Giraldez
University of La Laguna
Basic Medical Sciences
Campus Ciencias de la Salud s/n
La Laguna 38071
Spain

Re: 202513799

Dear Teresa,

Thank you for submitting your manuscript, entitled "GRIN2B disease-associated mutations disrupt the function of BK channels and NMDA receptor signalling nanodomains" to JGP. Your manuscript has now been seen by 3 reviewers, whose comments are appended below. You will see that the reviewers were very enthusiastic about the study and its potential impact and raised only minor concerns that should nevertheless be addressed prior to further consideration of the manuscript at JGP.

We hope that you will be able to submit a revised manuscript that addresses these points, which we believe will pose no problems, and which may be re-reviewed. In addition, please do not hesitate to contact me (via the editorial office) if you feel that a discussion of the reviewers' and editors' comments would be helpful.

Please submit your revised manuscript via the link below, along with a point-by-point letter that details your response to the reviewers' and editors' comments, as well as a copy of the text with alterations highlighted (boldfaced or underlined). If the article is eventually accepted, it would include a 'revised date' as well as submitted and accepted dates. If we do not receive the revised manuscript within one year, we will regard the article as having been withdrawn. We would be willing to receive a revision of the manuscript at a later time, but the manuscript will then be treated as a new submission, with a new manuscript number.

Please pay particular attention to recent changes to our instructions to authors in the following sections: Data presentation, Blinding and randomization and Statistical analysis, under Materials and Methods, as shown here: <https://rupress.org/jgp/pages/submission-guidelines#prepare>. Re-review will be contingent on inclusion of the required information (including for data added during revision) and demonstration of the experimental reproducibility of the results. Also, To improve the reproducibility of published content, we have partnered with SciScore. Authors are prompted in eJP to copy and paste the Materials and Methods section of their manuscript for a SciScore assessment when submitting their revised manuscript. Authors are encouraged (not required) to further revise their Materials and Methods if the SciScore is below 4. More information can be found here: <https://rupress.org/jgp/pages/submission-guidelines#sciscore>.

Please note, JGP now requires authors to submit Source Data used to generate figures containing gels and Western blots with all revised manuscripts (when applicable). This Source Data consists of fully uncropped and unprocessed images for each gel/blot displayed in the main and supplemental figures. If your paper includes cropped gel and/or blot images, please be sure to provide one Source Data file for each figure that contains gels and/or blots along with your revised manuscript files. File names for Source Data figures should be alphanumeric without any spaces or special characters (i.e., SourceDataF#, where F# refers to the associated main figure number or SourceDataFS# for those associated with Supplementary figures). The lanes of the gels/blots should be labeled as they are in the associated figure, the place where cropping was applied should be marked (with a box), and molecular weight/size standards should be labeled wherever possible. Source Data files will be made available to reviewers during evaluation of revised manuscripts and, if your paper is eventually published in JGP, the files will be directly linked to specific figures in the published article.

Source Data Figures should be provided as individual PDF files (one file per figure). Authors should endeavor to retain a minimum resolution of 300 dpi or pixels per inch. Please review our instructions for export from Photoshop, Illustrator, and PowerPoint here: <https://rupress.org/jgp/pages/submission-guidelines#revised>

Whilst you are revising your manuscript, we ask that you consider whether you have any artwork that might be suitable for the cover of JGP. Microscopy images are particularly good for cover artwork, but other types of image can be very effective, so we encourage you to be creative. Please don't restrict yourself to images from the paper; an image that is relevant to the work described would be just as suitable. Images should be a minimum resolution of 300 dpi. To see recent examples, visit the following page and click on 'Show covers? Yes': <https://jgp.rupress.org/content/by/year>

Thank you for submitting your interesting research to JGP.

Please submit your revised manuscript, and any associated files, via this link:
Link Not Available

Sincerely,

Crina Nimigean, Ph.D.
On behalf of Journal of General Physiology

Journal of General Physiology's mission is to publish mechanistic and quantitative molecular and cellular physiology of the highest quality; to provide a best-in-class author experience; and to nurture future generations of independent researchers.

Reviewer #1 (Comments to the Authors):

The ms by Martínez-Lázaro et al examines the role of functional interaction between BK channels and NMDA receptors. This is an extension of previous work by the authors' lab and other groups on functional complexes formed by NMDA receptors and BK channels, which may underlie a negative-feedback mechanism to limit neuronal excitability. Here the authors study two disease-linked GluN2B variants, V15M and V618G, to determine the potential molecular basis for functional deficits. The authors find that both variants on their own (without BK coexpression) yield NMDA currents that are essentially indistinguishable from WT. Whereas coexpression of NMDAR's with BK channels yielded large outward currents with application of 1mM glutamate with the WT NMDAR, indicating BK current that was coupled to NMDAR-mediated Ca influx, the outward current was diminished with the V15M variant and even more greatly diminished with V618G. These results are consistent with disruption in some aspect of BK activation by the receptors. The decreased current was not related to loss of Ca-permeability of the NMDAR variants. The V618G variant exhibited decreased functional coupling to BK channel activation based on electrophysiological measurements, despite robust surface expression. On the other hand, V15M receptors showed substantially decreased surface expression. Both variants exhibited decreased protein-protein interactions with BK channels, as indicated by proximity ligation assay and analysis of superresolution microscopy images.

The manuscript is generally well written and well organized, and the data are of good quality. This work falls well within the scope of JGP and should be of interest to the readership. It would be nice if the authors could provide some insight toward the physical mechanism of coupling, or the basis of its disruption by the mutations, but I imagine this may be left to a future study. I have a few comments for the authors to address:

- 1) The location of residue V15 within the GluN2B subunit is quite unclear to me. The authors mention that it is located on a "signal peptide" which I assume is not visible in available electron density maps, but I think some more information should be included in the manuscript, e.g. is V15 expected to be in a solvent-exposed region, is it near the transmembrane helices, or near the ligand-binding domain, etc.
- 2) It may not be obvious to some readers before reading the ms that GRIN2B is the name of the gene that encodes the GluN2B subunit, but GRIN2B is used in the title, so it certainly should be defined up front. However, the name of the GRIN gene family is not mentioned in the Introduction until line 130. Maybe the GRIN gene name should be introduced/defined at the same point in the Introduction where the authors first introduce the NMDA receptor subunits as GluN2B, etc. (i.e. around lines 81-82). Also, GRIN should be noted as the gene family encoding GluN subunits in the abstract (again for readers who are unfamiliar with the GluR gene names).
- 3) In Fig 2B, the outward currents are reduced when BK channels are co-expressed with the NMDA variants. Can the authors provide measurements of the voltage-activated BK current independent of glutamate application in these cells? I'm just wondering if there is evidence that co-expression of the NMDA variants isn't somehow reducing BK expression levels (or maybe I have missed this evidence). The authors should provide some comment on this.

Reviewer #2 (Comments to the Authors):

In this manuscript, Martínez-Lázaro et al. studied the impact of two mutations in the GRIN2B gene associated to neurodevelopmental disorders in NMDA receptor localization and NMDA receptor-BK channel functional coupling. The authors deployed an array of electrophysiological, biochemical, and imaging approaches, and found that the V15M mutation strongly decreases GluN2B expression, while the V618G mutation affects functional coupling. Overall, this manuscript provides important insights onto the functional consequences of clinically relevant mutations in NMDA receptor function, specifically in the context of the functional coupling with BK channels. The manuscript is clearly written, and data are well presented. I have a number of suggestions and relatively minor concerns, mainly concerning the biochemical studies and the conclusions that have been drawn from them, but these do not diminish my interest and support for the publication of the paper in the Journal of General Physiology.

1) There seems to be significantly less protein loaded in the mutant conditions (both input and surface) in the representative WB in Fig. 5. Additionally, lane width seems different in input vs surface samples.

I encourage the authors to provide new representative WBs for this experiment, where loading controls are comparable across samples. Alternatively, it would be interesting to see if Grin2BV15M / Grin2BV618G messenger expression is altered by qPCR.

It is possible that mutant GluN2A subunit expression compromises the viability of transfected HEK cells. Have the authors noticed a decreased viability of GluN1-GluN2BV15M or GluN1-GluN2BV618G-transfected cells (by general appearance of the culture, or total protein level comparison)? If this is the case, a discussion is warranted, including possible its possible confounding effects in other experiments, including the PLA presented in F7.

2) Related, why is PLA signal presented as relative to surface area (thus subject to confounds such as cell density or transfection efficacy), instead of GFP signal / number of transfected cells?

3) The authors used TIRFM to assess expression of WT and mutant GluN2B subunits at the plasma membrane. I am confused, however, by the representative images provided, which show that most of the GFP signal apparently originates from intracellular membranes. Please comment.

Similarly, I find the apparently homogeneously distributed GluN2B and BK signal in low magnification STORM images (Fig. 8a) surprising: wouldn't signal be expected to be enriched at the membrane? Please clarify.

Reviewer #3 (Comments to the Authors):

Martinez-Lazaro et al investigate the effects of the GluN2B subunit of NMDA receptors on the coupling between NMDAR and BK channels, which form macromolecular associations found in neurons across various brain regions in the CNS. Since the activation of BK channels dampens the excitatory responses mediated by NMDAR, the disruption of this complex could lead to dramatic changes in neuron excitability. In this study, the authors specifically examine the impact of two GluN2B mutations associated with neurodevelopmental disorders (V15M and V618G) on the ability of NMDAR to associate with BK channels. Their findings reveal that V15M mutation significantly reduces the expression of NMDAR at the plasma membrane, leading to fewer NMDAR-BK channel associations. In contrast, the V618G mutant disrupts the formation of functional NMDAR-BK macrocomplexes. The manuscript is clearly written, the experimental design is careful, and the data presented are of high quality. The conclusions drawn are well supported by the findings.

I only have one major concern that I believe could be addressed by the authors in a timely manner:

In the assessments of plasma membrane expression for GluN1-GluN2B(V618G), GluN1-GluN2B(V15M), and GluN1-GluN2B(WT) by biotinylation/western blot (Fig. 5) and TIRFM (Fig.6), the cell transfections did not include BK channels. Under these transfection conditions, the membrane abundance of different NMDAR combinations (significantly increased for V618G and decreased for V15M) seems to contradict the results in Fig. 2A (showing similar current densities for all NMDAR combinations alone). However, when BK is cotransfected with the different NMDAR combinations, all transfections result in similar calcium entry (Fig. 3). Could you discuss this apparent contradiction?

Minor detail:

In the Fig.2 Legend, panel C description: it would help the reader to specify that all plots refer to NMDAR/BK currents.

We would like to sincerely thank all reviewers for their positive comments and constructive criticisms on our work. We believe that responding to these concerns made our manuscript stronger. The reviewers' comments are shown in black; our responses are in blue.

Reviewer #1 (Comments to the Authors):

(...)The manuscript is generally well written and well organized, and the data are of good quality. This work falls well within the scope of JGP and should be of interest to the readership. It would be nice if the authors could provide some insight toward the physical mechanism of coupling, or the basis of its disruption by the mutations, but I imagine this may be left to a future study.

We sincerely thank the reviewer for the overall positive evaluation of our work and for the detailed comments and suggestions. Our responses are below.

1) The location of residue V15 within the GluN2B subunit is quite unclear to me. The authors mention that it is located on a "signal peptide" which I assume is not visible in available electron density maps, but I think some more information should be included in the manuscript, e.g. is V15 expected to be in a solvent-exposed region, is it near the transmembrane helices, or near the ligand-binding domain, etc.

We appreciate the reviewer's comment and agree that this point required further clarification. As with many transmembrane proteins, NMDAR subunits contain an N-terminal signal peptide that directs the nascent polypeptide to the endoplasmic reticulum during translation. In both rat and human GluN2B sequences (RefSeq NP_036706.1 and NP_001400921.1, respectively), this signal peptide encompasses the first 26 amino acids, including residue V15. This segment is cleaved off in the mature protein and is therefore absent from resolved NMDAR structures such as PDB: 7SAA, which is why V15 is not visualized in our structural representation (Figure 1 legend). Previous studies have also identified pathogenic variants in the signal peptide of GluN2B, such as V18I (Hu et al., 2016; PMID: 27818011). More broadly, recent analyses have shown that signal peptide mutations across the human genome can affect protein targeting, translocation, and stability (e.g., Gutierrez Guarnizo et al., 2023; PMID: 37859801). Our data support the interpretation that the V15M mutation disrupts GluN2B stability and membrane targeting, leading to reduced membrane expression. We have now included this explanation in the revised manuscript (Discussion, line 539) for clarity: *"This reduction is most likely attributable to the position of the V15 residue within the signal peptide at the extreme N-terminus, which directs the nascent GluN2B protein to the plasma membrane. In both rat and human GluN2B sequences (RefSeq NP_036706.1 and NP_001400921.1, respectively), the signal peptide comprises the first 26 amino acids and is cleaved during protein maturation, which explains its absence from resolved structural models. Beyond GluN2B, a genome-wide analysis of pathogenic signal peptide variants has shown that such mutations can impair protein targeting, translocation, processing, and stability (Gutierrez Guarnizo et al., 2023), consistent with our findings. Although other GRIN2B pathogenic variants*

within the signal peptide—such as V18I—have been reported (Hu et al., 2016), to our knowledge, this study presents the first detailed functional characterization of the V15M mutation. Our results suggest that its likely pathogenic effect stems from reduced protein expression and impaired membrane localization”.

2) It may not be obvious to some readers before reading the ms that GRIN2B is the name of the gene that encodes the GluN2B subunit, but GRIN2B is used in the title, so it certainly should be defined up front. However, the name of the GRIN gene family is not mentioned in the Introduction until line 130. Maybe the GRIN gene name should be introduced/defined at the same point in the Introduction where the authors first introduce the NMDA receptor subunits as GluN2B, etc. (i.e. around lines 81-82). Also, GRIN should be noted as the gene family encoding GluN subunits in the abstract (again for readers who are unfamiliar with the GluR gene names).

We thank the reviewer for this valuable suggestion. In the revised manuscript, we have now introduced the reference to the GRIN2B gene earlier in the Introduction (lines 66–67) to ensure clarity for readers unfamiliar with the gene–protein nomenclature. Additionally, we have explicitly clarified that the GRIN gene family encodes GluN subunits when first describing NMDA receptor subunit composition (lines 81–84). As recommended, we also updated the abstract: the GRIN gene family is now identified as encoding GluN subunits (line 27), and we confirm that GRIN2B is specifically stated to encode the GluN2B subunit (line 34). We hope these changes improve accessibility and clarity for all readers.

3) In Fig 2B, the outward currents are reduced when BK channels are co-expressed with the NMDA variants. Can the authors provide measurements of the voltage-activated BK current independent of glutamate application in these cells? I'm just wondering if there is evidence that co-expression of the NMDA variants isn't somehow reducing BK expression levels (or maybe I have missed this evidence). The authors should provide some comment on this.

We thank the reviewer for raising this point. While we do not have a sufficient number of recordings of voltage-activated BK currents in the absence of glutamate application to definitively compare BK functional expression across all conditions, we have indirect evidence supporting the notion that BK expression is not significantly altered by co-expression with different NMDAR variants. Specifically, our STORM super-resolution imaging experiments (Supplemental Fig. S1) show that the total fluorescent signal corresponding to BK labeling is highly comparable across all conditions, regardless of the NMDAR variant expressed. This suggests that BK protein levels and localization at or near the membrane remain consistent, and that the differences observed in glutamate-evoked outward currents likely reflect altered functional coupling rather than reduced BK expression. We have added the following clarification to the revised manuscript (line 635): “Another factor to consider would be the possibility that co-expression of the NMDA variants is affecting BK expression levels. STORM labeling did not support this idea, as the total BK fluorescent signal was comparable across all tested conditions (Supplemental Fig. S1).” We hope this explanation helps clarify the basis for our interpretation.

Reviewer #2 (Comments to the Authors):

In this manuscript, Martínez-Lázaro et al. studied the impact of two mutations in the GRIN2B gene associated to neurodevelopmental disorders in NMDA receptor localization and NMDA receptor-BK channel functional coupling. The authors deployed an array of electrophysiological, biochemical, and imaging approaches, and found that the V15M mutation strongly decreases GluN2B expression, while the V618G mutation affects functional coupling. Overall, this manuscript provides important insights onto the functional consequences of clinically relevant mutations in NMDA receptor function, specifically in the context of the functional coupling with BK channels. The manuscript is clearly written, and data are well presented. I have a number of suggestions and relatively minor concerns, mainly concerning the biochemical studies and the conclusions that have been drawn from them, but these do not diminish my interest and support for the publication of the paper in the Journal of General Physiology.

We are grateful to the reviewer for these positive comments and the careful analysis of our work. Below are our responses.

1) There seems to be significantly less protein loaded in the mutant conditions (both input and surface) in the representative WB in Fig. 5. Additionally, lane width seems different in input vs surface samples. I encourage the authors to provide new representative WBs for this experiment, where loading controls are comparable across samples. Alternatively, it would be interesting to see if Grin2BV15M / Grin2BV618G messenger expression is altered by qPCR.

We thank the reviewer for this important observation. We fully agree that the Western blot (WB) originally presented in Fig. 5 may have caused confusion. We are truly grateful for this comment, as it prompted us to carefully revise the Figure, resulting in clarifications and improvements that we believe enhance its quality and clarity. In addition, we recognize the need to provide further insights and technical details to reassure the reviewer and readers that the integrity of our data remains intact, and that the conclusions drawn from these experiments are fully supported.

First, we would like to inform the reviewer that, inadvertently, we included in the final version of the manuscript a superimposed WB image with overlaid marker lanes in color, resulting in an apparently overexposed blot. We sincerely apologize for this error. To correct it, in the revised manuscript we have now replaced the WB shown in Fig. 5 with the correct image. Importantly, we would like to clarify that the correct (non-overexposed) image was the one used for quantification in all cases. To further clarify this point, we present below (*Reviewer-only figure 1*) a direct comparison between the previously overexposed images and the actual WB used for quantification.

Second, while we understand the reviewer's concerns regarding the apparent differences in protein loading, we respectfully argue that this does not impact the validity of our conclusions, for the following reasons:

1. All data were normalized, specifically to mitigate any potential bias arising from variations in protein loading.

2. The total amount of protein in the biotinylated fraction is expected to be lower than that of total protein extracts, and this difference does not affect the normalization process applied. However, it does affect the size and appearance of the band, resulting in some cases in band shrinking. As the reviewer can see if the Figure provided, other gels show this effect to a similar extent. We chose the gel in Figure 5 because it best represents the average results.

3. Consistency in quantification across independent experiments was carefully maintained, as reflected in the low variability observed in the data (Figure 5B-C).

4. Finally, it should be considered that, as stated in the manuscript, the WB data are only semi quantitative. The main message that these data aim to convey, taking into account the experimental limitations of this technique, is that the membrane abundance of the mutant GluN2B-V618G is relatively higher than the WT, while that of the mutant CluN2B-V15M is relatively lower. We have revised the manuscript text to ensure that this aspect is well clarified when considering the interpretation of the WB data.

We hope that these clarifications, together with the corrected Figure, address the reviewer's concerns and reinforce the robustness of our findings.

Reviewer-only figure 1: Images correspond to two of the replicas of western blots of total and membrane GluN2B protein abundance in HEK293T cells non-transfected (NT) or transfected with GluN1-GluN2BWT (WT), GluN1-GluN2BV15M (V15M) or GluN1-GluN2BV618G (V618G). Input (I): total lysate abundance; Surface (S): biotinylated fractions corresponding to membrane proteins. Na⁺/K⁺ ATPase α 1 Subunit (ATP1A1) and total tubulin were used as reference (see Figure 5 and further description in the methods section, in the manuscript text). Arrows indicate migration of

molecular mass markers (MW; values in kDa). The images on the left correspond to the image obtained with the Optimal Auto-exposure function of the BioRad imaging system, which ensures the use of the full dynamic range of detection without saturation. The images at the right correspond to the merging of the colorimetric image of pre-stained protein markers with the chemiluminescent image of the same blot obtained after the maximum exposure time, therefore resulting in overexposition. Note the bands corresponding to the membrane fraction (S) in Rep#2. We chose the Rep#1 because the general quality of the WB is better and more representative of the data average.

It is possible that mutant GluN2A subunit expression compromises the viability of transfected HEK cells. Have the authors noticed a decreased viability of GluN1-GluN2BV15M or GluN1-GluN2BV618G-transfected cells (by general appearance of the culture, or total protein level comparison)? If this is the case, a discussion is warranted, including possible its possible confounding effects in other experiments, including the PLA presented in F7.

It has been reported that expression of NMDARs in heterologous expression systems is especially challenging due to the higher permeability to Ca^{2+} and activation by residual glutamate and glycine in the culture medium, which lead to Ca^{2+} influx into the cells, causing cytotoxicity and cell death (Chazot et al 1999, PMID 16113964). Co-expression of NMDAR with BK is therefore tricky, and the cells are not always as healthy as they are with other combinations of channels. In our lab, a great effort has been invested in optimizing the culture conditions. Our results showed that optimal conditions for NMDAR-BK expression must include: (1) transfection in OptiMEM + 5% FBS with low amounts of plasmid; (2) replacing the media 4 h after transfection with the same medium complemented with 200 μM DL-2-amino-5-phosphonovaleric acid (D-APV) and 20 μM 5,7-Dichlorokynurenic acid (DCKA), we culture the cells in the presence of NMDAR inhibitors. In these conditions, we did not observe consistent differences in viability or total protein levels between coexpression of the different NMDAR subunits (GluN2B) and/or mutants. In the new version of the manuscript, we now state this fact in the results section (line 440): *“It is important to note that we did not observe consistent differences in the viability of GluN1-GluN2BV15M or GluN1-GluN2BV618G-transfected cells (by general appearance of the culture, or total protein level comparison), so we discarded this as a possible factor to explain our results”*. Moreover, we are specially grateful to the reviewer for bringing up this comment since it made us realize that the methodological details explained above were not included in the methods section. They are now included in the revised version of the manuscript (lines 171-175).

2) Related, why is PLA signal presented as relative to surface area (thus subject to confounds such as cell density or transfection efficacy), instead of GFP signal / number of transfected cells?

We thank the reviewer for this question and the opportunity to clarify. In the PLA experiments, we are not quantifying GFP signal. The PLA signal corresponds to the number of red fluorescent puncta per cell, each indicating a close proximity (<40 nm) between the proteins of interest. These red puncta are automatically quantified and

normalized to the estimated cytoplasmic area of each individual cell, not to total area of the imaging field or to GFP signal. We agree that variability in transfection efficiency and cell size could be potential confounding factors. To minimize this, the PLA analysis software identifies nuclei and estimates cytoplasmic area on a per-cell basis, enabling normalization at the single-cell level. This approach, applied across >30–40 cells per condition and repeated in at least three independent experiments, reduces variability due to cell density, size, or transfection heterogeneity. Based on the reviewer's helpful suggestion, we have now revised the description of the PLA methodology in the Methods section to clarify these points for the reader (line 208): *"Quantification was performed at the single-cell level: nuclei were automatically identified, cytoplasmic area estimated for each cell, and PLA signal (red puncta) normalized to individual cell area. This per-cell normalization accounts for variability in cell size and transfection efficiency and was applied consistently across all experimental replicates"*.

3) The authors used TIRFM to assess expression of WT and mutant GluN2B subunits at the plasma membrane. I am confused, however, by the representative images provided, which show that most of the GFP signal apparently originates from intracellular membranes. Please comment.

We thank the reviewer for raising this point. We apologize for any confusion caused by the appearance of the TIRFM images. We believe the reviewer may have interpreted the signal as intracellular because it is not localized to the cell periphery, as typically seen in confocal microscopy. However, Total Internal Reflection Fluorescence Microscopy (TIRFM) selectively excites fluorophores within a very narrow evanescent field—approximately 100–200 nm from the coverslip—thereby illuminating primarily the basal plasma membrane in contact with the glass surface. Therefore, fluorescent signals—such as those from GluN2B-GFP—appear as a more uniform distribution across the basal footprint of the cell, rather than as edge-localized patterns. This is expected and distinct from the appearance of membrane labeling in confocal sections. That said, we acknowledge that TIRFM is subject to certain limitations, particularly in objective-based single-angle setups like ours. Irregular illumination can result from interference fringes, scattering, or shadowing by cellular structures, especially in regions with uneven topography (e.g. Ellefsen et al., 2015; PMID: 26308212). In our experience, partial detachment of HEK293T cells at room temperature can also contribute to this effect. To clarify this point for readers, we have revised the Methods section (line 302) to better explain the imaging characteristics and limitations of TIRFM in this context: *"TIRFM selectively excites fluorophores within ~100–200 nm of the coverslip, capturing fluorescence from the basal plasma membrane. This produces a more uniform signal across the cell footprint, which contrasts with the peripheral signal seen in confocal images. However, shadowing or uneven illumination may still occur due to interference patterns or irregular cell topography (Ellefsen et al., 2015)."*

Similarly, I find the apparently homogeneously distributed GluN2B and BK signal in low magnification STORM images (Fig. 8a) surprising: wouldn't signal be expected to be enriched at the membrane? Please clarify.

We thank the reviewer for this observation. The apparently homogeneous distribution of GluN2B and BK signal in Fig. 8A is due to the imaging configuration used. In our setup, STORM was performed in combination with TIRFM. For this reason (see also our response to the previous comment), the images primarily represent the basal plasma membrane of the cell—the surface in contact with the glass—rather than membrane edges or intracellular compartments. This yields a planar, footprint-like view of membrane-associated proteins (and, in this case, protein complexes), which may appear homogeneously distributed at low magnification. We have now clarified this point in the Methods section to avoid confusion (line 314): “Fluorescence excitation was limited to the basal ~200 nm of the cell using TIRFM-based illumination. This setup ensures that STORM images primarily represent the plasma membrane region adjacent to the coverslip. Thus, fluorescent signals from GluN2B and BK channels appear as a uniform distribution across the basal surface rather than lateral membrane edges”.

Reviewer #3 (Comments to the Authors):

Martinez-Lazaro et al investigate the effects of the GluN2B subunit of NMDA receptors on the coupling between NMDAR and BK channels, which form macromolecular associations found in neurons across various brain regions in the CNS. Since the activation of BK channels dampens the excitatory responses mediated by NMDAR, the disruption of this complex could lead to dramatic changes in neuron excitability. In this study, the authors specifically examine the impact of two GluN2B mutations associated with neurodevelopmental disorders (V15M and V618G) on the ability of NMDAR to associate with BK channels. Their findings reveal that V15M mutation significantly reduces the expression of NMDAR at the plasma membrane, leading to fewer NMDAR-BK channel associations. In contrast, the V618G mutant disrupts the formation of functional NMDAR-BK macrocomplexes. The manuscript is clearly written, the experimental design is careful, and the data presented are of high quality. The conclusions drawn are well supported by the findings.

We sincerely thank the reviewer for analysing our work and providing these positive comments.

I only have one major concern that I believe could be addressed by the authors in a timely manner: In the assessments of plasma membrane expression for GluN1-GluN2B(V818G), GluN1-GluN2B(V15M), and GluN1-GluN2B(WT) by biotinylation/western blot (Fig. 5) and TIRFM (Fig.6), the cell transfections did not include BK channels. Under these transfection conditions, the membrane abundance of different NMDAR combinations (significantly increased for V818G and decreased for V15M) seems to contradict the results in Fig. 2A (showing similar current densities for all NMDAR combinations alone). However, when BK is cotransfected with the different NMDAR combinations, all transfections result in similar calcium entry (Fig. 3). Could you discuss this apparent contradiction?

We thank the reviewer for this important observation. We agree that the differences in membrane expression observed in Figs. 5 and 6, when considered alongside the

current recordings in Fig. 2A and Ca²⁺ imaging data in Fig. 3, may appear contradictory at first glance. We believe this apparent discrepancy stems in part from the focus of the study, which is on assessing NMDAR-BK coupling efficiency rather than providing a full biophysical characterization of the isolated NMDAR mutants. This focus influenced our presentation choices, particularly in Figs. 2 and 3, where representative recordings were selected to highlight coupling differences under conditions of similar NMDAR inward currents. This approach may have inadvertently downplayed underlying differences in expression levels.

We also note that:

- The current traces in Fig. 2B are not normalized to cell size (i.e., not expressed as pA/pF), and thus cannot be directly interpreted as a measure of current density or membrane expression.
- Coupling efficiency, the key readout in this context, is computed as a relative measure ($Q_{\text{outward}}/Q_{\text{inward}}$), which inherently accounts for differences in NMDAR activity under each condition.
- For the V15M variant, we indeed encountered lower overall expression, which made it more difficult to obtain high-quality recordings. Despite this, when measurable, the coupling ratio remained comparable to WT.
- In Fig. 3, fluorescence signals ($\Delta F/F_0$) were normalized to baseline to emphasize that Ca²⁺ permeability per se is not compromised by either mutation.

Finally, as noted by the reviewer, we did not re-characterize the detailed biophysical properties of V618G, given the comprehensive prior work in this area (e.g., Lemke et al., 2014; Fedele et al., 2018; Vyklicky et al., 2018). Our goal was to build upon those findings to investigate the novel aspect of NMDAR-BK coupling.

We appreciate the reviewer's suggestion, which allowed us to improve the clarity of our presentation and interpretation. We now include a clarifying paragraph in the Discussion section to address this point directly and help reconcile the expression and functional data (line 586): *"It is important to note that while our biotinylation and TIRFM experiments (Figs. 5–6) revealed significant differences in membrane abundance among GluN2B variants, these differences were not mirrored in the raw NMDAR current amplitudes presented in Fig. 2A. This apparent discrepancy likely stems from the fact that our study focused primarily on assessing NMDAR-BK coupling efficiency, rather than conducting a comprehensive biophysical comparison of the isolated NMDAR variants. Representative traces in Fig. 2B were selected to ensure comparable inward NMDAR currents, allowing for a clearer evaluation of coupling differences. Additionally, the coupling efficiency metric ($Q_{\text{outward}}/Q_{\text{inward}}$) is inherently relative, and thus reflects the functional outcome of both expression and interaction within each specific condition. Furthermore, current densities were not normalized to cell size (pA/pF), and therefore do not provide a quantitative assessment of surface receptor abundance. The fluorescence-based Ca²⁺ imaging in Fig. 3, presented as normalized $\Delta F/F_0$ values, similarly served to illustrate that all three NMDAR variants support Ca²⁺ influx, rather than quantify absolute permeation levels. Together, these design choices were guided by the central aim of the study—to evaluate how GRIN2B mutations affect NMDAR-BK*

nanodomain coupling—rather than to recharacterize the well-studied properties of the V618G variant or fully define those of V15M.”

Minor detail:

In the Fig.2 Legend, panel C description: it would help the reader to specify that all plots refer to NMDAR/BK currents.

Done.

May 28, 2025

Prof. Teresa Giraldez
University of La Laguna
Basic Medical Sciences
Campus Ciencias de la Salud s/n
La Laguna 38071
Spain

Re: 202513799R1

Dear Teresa,

I am pleased to let you know that your manuscript, entitled "GRIN2B disease-associated mutations disrupt the function of BK channels and NMDA receptor signalling nanodomains" is scientifically acceptable for publication in Journal of General Physiology. Formal acceptance will follow when it is modified in accordance with the referees' remarks and our editorial policies.

Please note items that need attention are listed at the bottom of this email (under 'manuscript formatting checklist') and on the attached marked-up pdf file. Please also be sure to include a letter addressing the reviewers' comments point-by-point (if applicable) and a copy of the text with alterations highlighted (boldfaced or underlined). Your manuscript should be a double-spaced MS Word file and include editable tables, if appropriate.

Please submit your final files via this link:
Link Not Available

Thank you for choosing to publish your research in JGP and please feel free to contact me with any questions.

Sincerely,

Crina Nimigean, Ph.D.
On behalf of Journal of General Physiology

Journal of General Physiology's mission is to publish mechanistic and quantitative molecular and cellular physiology of the highest quality; to provide a best in class author experience; and to nurture future generations of independent researchers.

Manuscript formatting checklist:

- MS Word document of text needed (including editable tables)
- MS Word document of supplemental text needed (including figure legends and editable tables)
- Brief Statement describing supplementary information needed, if applicable (in subsection at end of Materials & Methods)
- Figures created at sufficient resolution and in acceptable format (including supplemental if applicable). If working in Illustrator, we prefer .ai or .eps file format. If working in Photoshop please use 600dpi/1000dpi .tiff or .psd file format. Minimum resolution at estimated print size: Minimum resolution for all figures is 600 dpi. For figures that contain both photographs and line art or text, 600 dpi is highly recommended. Figures containing only black and white elements (line art, no color, and no gray) should be 1,000 dpi. Maximum figure size is 7 in wide x 9 in high (17.5 x 22.8 cm) at the correct resolution. <https://jgp.rupress.org/fig-vid-guidelines>
- Supplemental figures conforming to same guidelines as manuscript figures (noted above)
- If images resemble one from a prior publications, the author must seek permissions (to reproduce or adapt) from the original publisher. [You can resubmit your paper while waiting to hear back from the original publisher but please keep us updated]
- All authors must complete a disclosure form prior to acceptance. A link to complete the form has been sent to all coauthors. Please provide the editorial office with updated email addresses if necessary

Reviewer #1 (Comments to the Authors):

The ms by Martínez-Lázaro et al examines the role of functional interaction between BK channels and NMDA receptors. This is an extension of previous work by the authors' lab and other groups on functional complexes formed by NMDA receptors and BK channels, which may underlie a negative-feedback mechanism to limit neuronal excitability.

The authors have address previous critiques to my satisfaction and I have no further comments.

Reviewer #2 (Comments to the Authors):

In this revised manuscript, Martínez-Lázaro et al. addressed the reviewers' comments and suggestions, and provided revised images for WBs in Fig. 5. The author's arguments are convincing and the additional details provided help clarify the methods applied (and their limitations) and the results achieved. The manuscript is, in my opinion, ready for publication at the Journal of General Physiology.

Reviewer #3 (Comments to the Authors):

The revised version of the manuscript has addressed my concern.